

# In-flight performance of the Ozone Monitoring Instrument

V.M. Erik Schenkeveld[1], Glen Jaross[2], Sergey Marchenko[3], David Haffner[3], Quintus L. Kleipool[1], Nico
C. Rozemeijer[4], J. Pepijn Veefkind[1,5], Pieternel F. Levelt[1,5]

[1]Royal Netherlands Meteorological Institute KNMI, De Bilt, The Netherlands
[2]NASA Goddard Space Flight Center, Greenbelt, Maryland
[3]Science Systems and Applications Inc., Lanham, Maryland
[4]TriOpSys BV, Utrecht, The Netherlands
[5]Delft University of Technology, Delft, The Netherlands

*Correspondence to*: V.M.E. Schenkeveld (erik.schenkeveld@knmi.nl)

**Abstract.** The Dutch-Finnish Ozone Monitoring Instrument (OMI) is an imaging spectrograph flying on NASA's EOS Aura
satellite since July 15, 2004. OMI is primarily used to map trace gas concentrations in the Earth's atmosphere, obtaining
mid-resolution (0.4-0.6 nm) UV-VIS (264-504 nm) spectra at multiple (30-60) simultaneous fields of view. Assessed via
various approaches that include monitoring of radiances from selected ocean, land, ice and cloud areas, as well as
measurements of line profiles in the Solar spectra, the instrument shows low optical degradation and high wavelength
stability over the mission lifetime. In the regions relatively free from the slowly unraveling 'row anomaly' the OMI
irradiances have degraded by 3-8%, while radiances have changed by 1-2%. The long-term wavelength calibration of the
instrument remains stable to 0.005-0.020 nm.

## 1. Introduction

The Dutch-Finnish Ozone Monitoring Instrument (OMI) is an imaging spectrograph flying on board the NASA's EOS Aura
satellite since July 15, 2004. OMI is used to measure atmosphere trace gasses ($O_3$, $NO_2$, $SO_2$, HCHO etc.), aerosol
characteristics, and various other quantities (e.g., surface UV radiation). The instrument has previously been described in
Levelt et al. (2006), its calibration in Dobber et al. (2006) and the Level 1B data processor in Oord et al. (2006). OMI
continues to extend the 25-year record of ozone measurements that started with the SBUV (Cebula et al., 1988) and TOMS
(McPeters et al., 1998) instruments of the National Aeronautics and Space Administration (NASA). This record has been
continued by measurements from the Sciamachy (Bovensmann et al., 1999), GOME (Burrows et al., 1999) and GOMOS
(Kyrölä et al., 2004) instruments of the European Space Agency (ESA) and Eumetsat. Currently the OMPS instrument
(Seftor et al., 2013) of NASA is also performing ozone measurements. After end-of-life of OMI, the ozone record will be
continued by the future instruments Tropomi (Veefkind et al., 2012), Sentinel 4 (Bazalgette Courrèges-Lacoste et al., 2008)
and Sentinel 5 (Sierk et al., 2012) from ESA, GEMS from South Korea (Kim et al., 2015) and TEMPO from NASA (Chance
et al., 2013). Tropomi and Sentinel 5 will be in a polar, sun-synchronous orbit comparable to the OMI orbit. Sentinel 4,



TEMPO and GEMS will be in a geostationary orbit, enabling them to monitor regions of the Earth with high temporal (e.g. diurnal) resolution. It is estimated that the product results will be available every hour during daytime, whereas OMI, Tropomi and Sentinel 5 only have a product delivery once per day. The older instruments needed even more time to get global coverage. At the time of writing of this article OMI continued to provide high-quality science data. This article

focuses on the long-term OMI performance, primarily addressing results of in-flight calibration and changes in the instrument during the mission.

This article begins with a description of the instrument. The optical design is shown, followed by a description of the detector. The primary goal of the instrument is to measure Earth radiances and solar irradiances which are reported in level 1 products. The requirements for these level 1 products are described in section 1.1. The measured radiances and irradiances

are used by retrieval algorithms to estimate ozone, trace gasses as well as aerosol properties and UV irradiance. The results of the retrievals are stored in level 2 products. A description of these algorithms and the level 2 products is beyond the scope of this article. The instrument has shown a modest degree of degradation over time. The most important source of this degradation is that of the diffusers which are used to measure solar irradiances. The CCD detectors have also degraded over time, resulting in a higher dark current and an increasing number of bad pixels. Multiple instrument performance and health

parameters are monitored (semi-) continuously during the mission. Some parameters are measured in every orbit, while others are only measured once per day or per month. This monitoring results in trends of the parameter over the mission. Parameters that are important for calibration corrections and science product retrievals are discussed. The most important of these parameters are solar diffuser degradation, detector degradation and change in spectral calibration. A set of parameters that characterizes the so-called "row anomaly", it's evolution since June 2007, and it's impact on the measured radiances is

also shown.

### 1.1.  Basic goals and requirements for Level 1 products

The OMI spectrograph acquires mid-resolution (0.4-0.6 nm) spectra in the 264-504 nm wavelength range. This spectral region is measured by 3 instrument channels, UV1, UV2 and VIS (Table 1). The wide, 115-degree swath angle permits a ground track with a swath width of 2600 km. This broad swath is simultaneously imaged in 60 cross-track field-of-view

channels (detector rows) for the UV2 and VIS channels, and 30 for the UV1 channel. OMI follows a polar, sun-synchronous orbit with an average altitude of 705 km, orbital period of 98 minutes 53 seconds and an ascending node local time of 1.42 PM (note that the equator-crossing time was slightly adjusted during the mission). OMI orbits the Earth 14 times per day, providing daily global coverage in nominal operation. The short exposure times (typically, around 2 sec) result in a spatial resolution of 13 km in the flight direction.

There are 6 standard OMI Level 1 data products (Table 2). The products used to generate level 2 products are generally the global UV and VIS radiance products and accompanying irradiance data. The Spatial Zoom products are produced one day per month. These products zoom in on a smaller swath on ground with a higher spatial resolution. The irradiances are



measured once per day. The calibration product provides, for each orbit, the dark current, background, WLS and LED measurements and, whenever applicable (i.e., only at the times of solar calibration), the solar irradiance measurements in different formats, which supplements the standard irradiance output.

## 2. Instrument design

In this section a description of the optical (Figure 2 and Figure 3) and detector (Figure 4) design of the Ozone Monitoring Instrument (OMI) is presented. The Ozone Monitoring Instrument (Levelt et al., 2006 and Dobber et al., 2006) is a Nadir viewing imaging spectrograph where the UV and visible range of the Earth spectrum is imaged onto two CCD detectors. One dimension of each CCD detector is used for the wavelength measurement, and the other dimension is used for spatial measurement of the cross track field of view perpendicular to the flight direction. An impression of OMI flying over the Earth surface illustrates the flight and measurement configuration in Figure 1.

### 2.1. Optical design

The Earth radiances and solar irradiances are acquired through mostly identical optical pathways.

#### 2.1.1. Radiance channel

The Earth radiance is imaged via the telescope (primary and secondary mirror) onto the entrance slit. A polarization scrambler is placed in the vicinity of aperture stop and before the secondary mirror of the telescope. The secondary mirror projects images onto the entrance slit of the spectrograph. A dichroic mirror is placed behind the entrance slit and reflects the UV part of the radiance spectrum to the UV channel and transmits the VIS part of the spectrum to the VIS channel. The UV light passes a field lens and then the grating creates the image of the UV spectrum. A mirror splits the UV spectrum in two parts UV1 and UV2. The UV1 and UV2 channels are split, because the requirements for these channels are different. The UV1 channel is designed to detect the shortest wavelengths. This channel is primarily used for the detection of ozone profiles. The UV signal decreases rapidly for shorter wavelengths, resulting in a low signal to noise ratio (SNR). To increase the SNR a design choice has been made to increase the detector pixel size, at the expense of spatial resolution. The result is that only 30 spatial channels are available for UV1, versus 60 for UV2 and VIS. Also the choice of coatings on the optical elements is optimized for each spectral channel. Two sets of objective lenses project the spectrum onto the UV CCD detector. In the VIS channel a set of mirrors project the VIS signal onto the grating. The objective lenses project the image of the spectrum onto the VIS CCD detector.



### 2.1.2. Irradiance channel

In the irradiance channel the first component the sunlight passes is the optical mesh with 10% transmission. The sunlight can enter the instrument if the Solar Aperture Mechanism is opened. This is done once per day just before the spacecraft enters into eclipse, at the northern part of the Earth. The sunlight is then reflected by one of the reflection diffusers: Quartz Volume
Diffuser (QVD), regular Aluminum or backup Aluminum. These diffusers are mounted on a diffuser carousel. The QVD diffuser used in daily irradiance measurements. To monitor degradation of the QVD diffuser, the regular Al diffuser is used once per week and the backup Al diffuser once per month. After being reflected by the diffuser, the light can be reflected by the folding mirror (FM), once the mirror is moved to the sun-observing position, thus blocking the Earth light. After reflection by the folding mirror the optical path is identical for radiance and irradiance. Thus, the difference between the
radiance and irradiance optical pathways is the primary mirror for the radiance channel, and the reflection diffuser and folding mirror for the irradiance channel.

### 2.1.3. Calibration channel

OMI is equipped with two calibration light sources: a Quartz Tungsten Halogen (QTH) White Light Source (WLS) and green LEDs. The WLS is imaged via a lens and two mirrors onto a transmission diffuser, which is mounted on the diffuser
carousel. When the WLS is used for measurements, the Folding Mirror is placed in the calibration position. This position will block the Earth radiance. The WLS is used to measure changes in the CCD performance, in particular the pixel-to-pixel response non-uniformity. The WLS can also be used to monitor radiometric throughput. Both the UV and VIS channel are equipped with two green LEDs. These LEDs are placed just before the CCD detector. In the VIS channel the LED light passes directly through the channel objective; in the UV channel the illumination is indirect. The LEDs can be used to
monitor the CCD pixel behavior and linearity of the detector and electronics.

### 2.2.   Detectors and electronics unit

The OMI instrument is equipped with two CCD detectors and one Electronics Unit (ELU). The CCD detectors in the UV and VIS channel are back-illuminated UV-enhanced silicon-based CCDs. These detectors (see more details in Dobber et al. (2006)) have 780 (spectral; hereafter designated as column) x 576 (spatial, or row) pixels. The operational temperature of the
CCDs is 265 K, stabilized with a precision of +/- 10 mK. The ELU controls and reads out both CCD detectors. The detectors have basic read-out electronics and a programmable gain. The relative gain values can be G1, G4, G10 and G40 resp., and can be programmed for certain column (i.e., spectral) ranges, thus providing extended dynamic range, critically important in the UV spectral region affected by strong ozone absorption. Three different gain switch columns can be used per CCD detector. Figure 4 shows the UV CCD detector layout. The CCD detector is divided in different parts. The useful (ir)radiance
signal is detected by the UV1 and UV2 areas in the image area. Above and below are stray light areas, designated for





dynamic estimates of stray light levels. At the extreme ends (rows) of the image area are dark-current areas. During readout the entire image area is transferred to the storage area. From there the image is read out via the read-out register. During the read-out a new image is acquired in the image area. The VIS channel carries similar designated areas.

### 3. Calibration concept and implementation

Calibration measurements for OMI are performed every day. These comprise solar, background, dark-current measurements, as well as the data from dedicated on-board stimuli. The optical paths for radiance, irradiance and calibration measurements are almost identical, except for a few elements (Figure 5).

For radiance measurements both the primary and secondary mirror M1 and M2 of the telescope are used. For calibration measurements the folding mirror FM is put in the light path between M1 and M2, effectively blocking the Earth shine. Solar

light enters the instrument via the diffuser D in reflection mode. Calibration measurements with the internal white light source use almost the same configuration as the solar calibration. The only difference is that the internal calibration light passes through the diffuser in transmission mode. Radiance and calibration pathways comprise the same optical elements, except mirror M1 for radiance and the diffuser D and the folding mirror FM for calibration measurements. Thus, in general these calibration pathways are suitable for calibration and degradation monitoring of all optical elements except mirror M1.

### 15   3.1. Calibration data analysis

Below we discuss the results of analysis of the L1B and telemetric data performed in 3 different ways. The Trend Monitoring and Calibration Facility (TMCF, hosted by KNMI, the Netherlands, see TMCF (2006)), performs basic analysis of daily L1B and telemetry data. In the second approach we evaluate the widths, depths and wavelength positions of well-defined absorption features (usually, blends of spectral lines) in the solar and Earthshine spectra. Lastly, we analyze long-

term trends in the OMI radiances observed over various geographical areas, as well as relatively small changes in the daily irradiance measurements. In particular, we pay attention to a sub-set of data acquired over the ice fields of Greenland and Antarctica, i.e., the regions with relatively stable, spatially homogeneous and predictable reflectances.

### 3.2. Wavelength registration approach

There are two wavelength registration approaches used in the OMI radiances and irradiances. Hence, the L1B OMI products

provide two slightly different wavelength grids. Here we briefly summarize the algorithms, providing more detailed discussion in Appendix A.

During the pre-flight testing and characterization the wavelength calibration has been performed using a PtCrNeAr spectral line source (Dobber et al., 2006). Narrow wavelength windows were centered on prominent spectral lines with accurately known wavelengths: 5 lines in the UV1 channel, 9 UV2 and 9 VIS lines. In each window the observed line profile was fitted




(in the CCD pixel space) with: a Gaussian function in the UV1 channel, and a sum of a Gaussian and a flat top function in the UV2 and VIS channels. The fits provided 3 sets (one per channel) of line-peak positions which were fitted with a fourth-order polynomial and translated into wavelength grids, thus providing the wavelength value for a given CCD pixel (column number).

In-orbit wavelength assignment for radiances and irradiances is done using two methods. In the first approach the wavelength assignment is based on pre-launch and early in-orbit wavelength calibration parameters, i.e., the polynomial coefficients modified as a function of Optical Bench temperature. For the UV2 and VIS channel this function is corrected for wavelength shifts that result from inhomogeneous slit illumination (see more details in the Appendix A). The result is a wavelength map:

$$\lambda(i,j) = \sum_{n=0}^{N} c_{2,n}(j)\left(i - ii_{ref}\right)^n$$

(1)

where i is column number, j is row number, n is the number of polynomial coefficients (typically 4) and $ii_{ref}$ is a reference column for the spectral calibration polynomial coefficients. The wavelength coefficients c(j) and the reference column are stored in the L1B radiance and irradiance output products.

For the second method, the wavelength calibration is performed by fitting a reference solar spectrum (Dobber et al, 2008b), an ozone absorption spectrum and a Ring spectrum to measured radiances. The latter two components are excluded from the

irradiance fits. The reference spectrum is divided in 8 windows for the UV1 channel, 18 windows for the UV2 channel and 22 windows for the VIS channel. The fits provide a set of wavelengths that are approximated by a polynomial with the corresponding coefficients (similar to eq. 1) stored in the L1B calibration product. In the original design of the OMI L1B processor only the parameters based on the first method were stored in the L1B products. These are the standard wavelength calibration parameters predominantly used by Level 2 developers. Later in the mission the wavelength-fit parameters were

also recorded in the L1B calibration product. Users are advised to implement the wavelength parameters of the first method. Expert users may also benefit from the wavelength-fit method, once they find that these parameters are more suitable for a particular L2 application.

## 4. Basic instrument performance

The following chapter describes the basic performance of the OMI instrument during 12 years of flight. The basic

performance of OMI has been monitored using the Trend Monitoring and Calibration Facility (TMCF, 2006). We also developed various trending tools supplementing and extending the basic TMCF metrics.



### 4.1. Monitoring calibration light sources

Calibration measurements with the LED are performed once per day, and with the WLS once per week. In the analysis of this data the average signal in the OMI channel (UV1, UV2 or VIS) is divided by a reference signal, which is an average signal of that channel at the beginning of the mission. The detected long-term changes in the calibration light sources are

summarized in Figure 6 and Figure 7. These are summaries of the overall changes of the calibration pathway throughput, therefore it is not possible to distinguish between the degradation caused by the light sources, the optical elements, the detector or the electronic components. The WLS (Figure 7) shows three abrupt changes in years 2006, 2008 and 2009. This source is used once a week, usually being switched on only for a short duration. So far, three times the WLS was activated for about 14 minutes (cf. the routine 1-min long calibration cycles). Such long duty cycle causes a temperature spike inside

the WLS bulb, making the halogen cycle more effective. During these events the intervening tungsten depositories are removed from the inner surface of the bulb, thus increasing the WLS output. There is no explanation for the erratic WLS behavior starting in 2012.

The main purpose of the LEDs is to monitor linearity and CCD detector properties. The observed 15% decrease in the LED output over the mission time does not impede the calibration routines. The main purpose of the WLS is a monitoring of the

CCD detector properties, Pixel Response Non-Uniformity (PRNU) inclusive. This source is not used for radiometric calibration. An overall long-term decrease of 10 % in the lamp output, as well as the three abrupt increases of the lamp output do not pose any problems for the relevant CCD characterization.

### 4.2. Detector degradation

The OMI CCD detector is proven to be sensitive to cosmic radiation, despite the ~10 kg of aluminum shielding (Dobber et

al., 2006) added to the instrument pre-flight. While in orbit, the cosmic radiation causes 3 effects: very short-duration spikes; changes in dark current; increase in the numbers of pixels affected by random telegraph signal changes (see below). These three effects are accounted for by the ground-processing software. The short duration peaks are corrected with a transient outlier correction algorithm. The increase in RTS pixels is detected, and a flagging scheme is updated accordingly. To correct for an increase in dark current, background measurements are performed at the night side of every orbit. There is a

full-cycle analysis with quality control of these background measurements in the TMCF that results in a daily update of the time-dependent Operational Parameter File (OPF) used in the L1B processor. The parameters that have the most impact on level 2 data retrieval and parameters demonstrating long-term instrument stability are discussed in the following subsections.

### 4.2.1. Detector electronics: gain ratios and electronic offsets

The measured spectrum that comes out of the CCD detector is amplified by the Electronics Unit (ELU). Different parts of

the measured spectrum are assigned different electronic gains, thus substantially improving the data quality at the



wavelength affected by strong ozone absorption (UV1 and UV2 range in particular). The high-gain parts will then have less readout noise and quantization noise. Both CCD detectors can be divided in 4 different areas, each with its own gain setting. Absolute gains cannot be measured during the mission, but relative gains (the gain ratios) can. The gain ratios are calculated out of a series of LED measurements. Once per month 10 LED measurements are recorded for all 4 gains, immediately

followed by a series of 10 LED measurements with the gain G1 (gain factor equals 1). These 10-exposure series are averaged, then normalized by the G1 average. Then the 4 areas in the resulting image with different relative gain values are averaged. This results in 4 relative gain values. The ratio of the measurements with gain setting G1 should be exactly one. In the top left panel of Figure 8 we see that the ratio is 1.0002 for the VIS channel. This is an indication for the accuracy of the analysis method.

The variations in gain values are not corrected by the L1B processor, hence they should be accounted for as multiplicative errors of the output signal. The shown long-term UV gain changes (~0.05%-0.25%) should not be perceived by the majority of L2 algorithms that are usually sensitive to radiometric errors in excess of ~0.25%-0.5%, especially when errors show distinctive wavelength dependence. Such wavelength dependence does not exceed ~0.1-0.15% in the UV case. The changes are more noticeable (0.25-0.35%) in the VIS domain. However, the temporal behavior of the VIS gains is essentially (to

~0.1%) the same. Hence, this suggests a negligible impact on the L2 retrievals.

Every measurement (radiance, irradiance or calibration) has an electronic offset. The electronic offset is added to the signal to prevent negative values in the amplified signal. Each gain setting has a different offset value. The electronic offset is determined from the first read-out in a measurement (the read-out register). All read-out register measurements from the mission are stored in the TMCF. The electronic offsets are evaluated as follows. In a read-out register measurement, all

pixels with the same gain value are averaged. Since there is no signal in a read-out register measurement, this average equals to the electronic offset for a given gain value.

Behavior of all offset values during the mission is shown in Figure 9. We detect the largest variation around 0.5 %. This is accounted for as an additive error and corrected by the L1B processor. Therefore such changes do not impact level 2 retrievals. From Figure 9 it can be seen that the trend in the UV channel differs from the VIS channel. The two channels

have individual CCD detectors and supporting electronics, however of a similar design. Hence, both detectors should show similar temporal behavior. The registered differences remain unexplained. The shown trends are based on standard radiance products, where the gain settings G10 and G40 are not used for the VIS channel. Thus, the VIS data are lacking in the lower panels of Figure 9.

### 4.2.2. Linearity of the CCD output amplifier

The output amplifier of the CCD can cause significant non-linearity effects when the incoming signal produces more than 2e5 electrons (67 % of the pixel full well). All measurements (radiance, irradiance and calibration) are corrected for this non-linearity effect by the L1B processor. If signal exceeds the 2e5 electrons limit for a certain CCD pixel, a non-linearity flag





will be raised for that pixel. The percentages of pixels with non-linearity flags are shown in Figure 10. We regard the percentage of flagged pixels as reasonably low. It does not exceed ~0.1% and ~0.001% in the VIS and UV channels, respectively, with an overall tendency to gradual diminishing on a top of distinctive seasonal cycles for the former, and noisy behavior (albeit around a very low level) for the latter.

Once per month linearity measurements with the LED and WLS are performed. For the LED a series of binned measurements with exposure times between 0.1 and 6 seconds is done. For the WLS the exposure times are between 0.4 and 1.6 seconds. Analysis of the data has shown that the WLS measurements are not suitable for non-linearity analysis because the WLS shows too much drift during a measurement, up to 1.4 % where the total non-linearity is expected to be around 3 %. The drift of the LED during a measurement is smaller than 0.1 %. The linearity analysis results in curves of deviations

from linearity vs. register charge. These curves for a number of samples during the mission are shown in Figure 11. It can be seen that non-linearity does not vary much during the mission.

### 4.2.3. Detector dark currents

The OMI CCD detectors are operated at T=265 K, controlled to +/- 10 mK. At this temperature the dark current of the CCDs was 85 el / (pix.sec) for UV and 132 el / (pix.sec) for VIS at the beginning of the mission. Dark current measurements are

performed on the night side of each orbit, thus revealing the levels of CCD's shot noise. The dark-current level is directly proportional to exposure time, hence the night-side calibration employs various integration times, up to 136 sec duration. In the analysis we use image-averaged dark current measurements. These averages are corrected for electronic offset and divided by the integration time and provided for each orbit in the L1B calibration files. As expected, during the mission the average dark current has gradually (Figure 12) increased, to ~800 el / (pixel . sec). The CCD detector is sensitive to cosmic-

ray hits, that result in the observed dark-current increase. The GOMOS mission (Kyrölä et al., 2004) has used the same CCD detector. Analysis of dark current trends of that detector has shown an increase of dark current during the mission (Bertaux et al., 2010) that is even higher compared to OMI. Though seemingly substantial, such increase does not negatively reflect of the L2 data production, as long as the processing pipelines use dark-current measurements taken at each OMI orbit. These measurements are used to correct the (ir)radiance measurements from the same orbit, thus effectively removing the dark

current contribution.

The increase in dark current can also be seen in the dark current distribution. Histograms for the UV channel for various years are shown in Figure 13, along with the UV bad-pixel threshold (see section 4.2.5: when a pixel has a dark current value above the pre-set threshold, it is flagged as a bad by the L1B processor). The corresponding histograms for the VIS channel look similar.





### 4.2.4. Random Telegraph Signal

A pixel affected by Random Telegraph Signal (RTS) has an average dark current that randomly toggles between two or more levels. Hence, for a given pixel its RTS behavior can be deduced via statistical analysis of the corresponding dark-current levels. This analysis is been done on specific dark current measurements that are performed once a day, and employ long integration time in order to improve statistics. There are two different dark current measurements used in this analysis, with 136 and 2 sec integration times, acquired at the same orbit. The measurement with the short integration time is subtracted from the measurement with the long integration time. For each pixel in the resulting image the dark current is calculated by dividing the signal by the difference in exposure times. A series of measurements is taken, belonging to 60 consecutive days. From this dataset the following statistics are calculated for every pixel: mean, variance, Observed to Expected variance Ratio (OER), skewness and kurtosis. Each statistics has pre-set threshold values. If, for a given pixel one of the statistics exceeds the threshold, it is flagged as a RTS pixel. Examples of a few RTS pixels and their histograms are shown in Figure 14. The number of pixels that show RTS behavior has increased from 0.1% to 0.7 % over the mission time (Figure 15).

The results shown in Figure 15 are for unbinned pixels. Standard radiance and irradiance measurements are performed by binning 8 consecutive rows (cross-track direction). If one or more of the 8 original (unbinned) pixels carries the RTS warning flag, then the binned pixel will also carry the RTS warning. Therefore, in the L1B products the amount of RTS flagged pixels is 8 times higher than shown in Figure 15.

### 4.2.5. Bad pixel flagging

Pixels are considered bad if their behavior is perceived as off-nominal. This can, for instance, be related to the too high or too low dark current, or a response to illumination that is too large or too small. This is monitored in the TMCF using dark-current, WLS and LED measurements. If values exceed the absolute threshold limits during dark current measurement, the pixel is flagged as bad or dead. The limit values for flagging have been determined in the first years of the mission, in an empirical manner, mostly based on the dark current values at that time, as well as the notion that the number of pixels being flagged should be neither exceedingly high nor too low. The upper limit value for a bad pixel is 2000 el / (pix . sec) and for a dead pixel is 3000 el / (pix . sec). The lower-limit values have been set to 1 el / (pix . sec). These values have not been changed during the mission. Since the dark current values are steadily rising, so does the numbers of flagged pixels. By definition, a dead pixel does not necessarily have zero response to incoming light. It can also be called a hot pixel, because the dark current of the pixel is much higher than expected. If a pixel signal deviates too much from the average pixel value in a WLS or LED image, it is also flagged as bad or dead. The result of this analysis is a bad and dead pixel map, that is recorded in the Operational Parameter File (OPF) for further use in the L1B processor. Predominant percentage of bad pixel flags is related to the pixels surpassing the pre-set dark-current bad-pixel threshold value of 2000 el / (pix . sec). The bad pixel percentages for the UV and VIS channels are shown in Figure 16.





The trends in Figure 16 should be considered in conjunction with the results from Figure 12 and Figure 13. The dark current of the detectors is increasing (Figure 12), hence a larger number of pixels is flagged as bad. As in Figure 15, Figure 16 shows the unbinned pixels. Therefore, due to binning the amount of the flagged bad pixels in the L1B products is 8 times higher than shown in Figure 16.

**4.2.6. Signal to Noise Ratio**

Direct estimate of Signal to Noise Ratio (SNR) from radiances is a very challenging task, in absence of calibrated, stable on-board sources with known spectral output. This is compounded by earthshine variability stemming from ever-changing geophysical factors. Hence, in order to reveal SNR trends, we revert to the values provided by OMI irradiances as well as specific OMI science products derived from irradiances.

Each daily solar measurement comprises 77 individual exposures taken within a fairly narrow range of relative (to QVD) solar elevations. We use these measurements for SNR estimates by choosing all the data within ±3 degrees elevation range. For each wavelength step in each exposure and each FOV (row) we calculate the difference between the measured irradiance and the average of irradiances from adjacent wavelengths. Then the RMS values of these differences are binned over 11 wavelength steps and averaged over all rows. The resulting SNRs are shown in Figure 17, where full and dotted lines follow

the monthly averages for January 2005 and 2016, respectively. The small, but systematic SNR decline goes in line with the steady growth of RTS events.

Some of the science products turn out to be more sensitive to the gradual SNR decrease. As an example, Figure 18 shows time-binned (yearly) RMS of the ratios of the solar indices measured in the VIS (Ca II lines) and UV1 (Mg II line) irradiances measured by OMI (Deland and Marchenko, 2013) and SORCE (Snow et al., 2005) on a daily basis. For a

particular spectral line (usually, the prominent absorption lines, such as MgII at 280 nm, or H and K CaII at 393, 397 nm), the solar index is defined (see more details in Deland and Marchenko (2013)) as a ratio between the solar flux at the line core to average solar flux measured at line's wings. Such indices serve as very sensitive indicators of solar activity, with the lines in question, MgII and CaII, changing in almost perfect agreement, however with the line-profile variability in the CaII lines being, on average, 5-7 times lower than the relative changes in the MgII lines. Inspection of the solar MgII indices provided

by OMI and, independently, by SORCE shows no discernible time-dependent trends in the relative noise level (Deland and Marchenko, 2013). This is not surprising, considering the relatively small UV1 SNR changes seen in Figure 17. Hence, we regard the MgII values as a relatively noise-free baseline and conclude that the gradual growth of RMS in the ratio of the CaII and MgII indices is caused by a steadily increasing instrument noise, with OMI CaII data being highly susceptible to these changes.



### 4.2.7. Pixel Response Non-Uniformity

Individual CCD pixels respond differently to incoming light. This is a detector property that depends on wavelength. This Pixel Response Non-Uniformity (PRNU) is about 5 % for wavelengths around 270 nm, and decreases to 0.1 % for wavelengths around 500 nm. If PRNU is not corrected properly, it will cause high-frequency structures in the calibrated L1B output products. To determine PRNU a white light source with a high spectral stability (up to $10^{-4}$) is needed. In practice it is difficult to obtain such a light source, both on-ground and in flight. On ground the PRNU has been determined by illuminating the QVD diffuser, because this diffuser introduces the smallest features (around $10^{-4}$). In flight the only option is to use the WLS light that passes through the transmission diffuser, which introduces features on a 1 % scale. The transmission diffuser feature pattern has been determined by comparing an on-ground measurement with the QVD diffuser and an on-ground measurement with the WLS light through the transmission diffuser (Dobber et al., 2006). Still it turned out to be very difficult to correct in-flight WLS measurements for the transmission diffuser features. The WLS is degrading in flight, and the dark current and noise of the CCD detector are increasing. Therefore the PRNU evolution during the mission has not been monitored. The PRNU correction in the L1B processor is currently still based on the results of the on-ground calibration.

### 4.3. Stray light

Various OMI L2 products show different sensitivity to the stray light contamination, depending on how the OMI radiances and irradiances are combined in a specific product. Below we show that in most cases any long-term trends that may be ascribed to gradual changes in the stray light levels do not exceed the achieved ~0.5%-1.0% detectability limits. One should note, however, that some spectral domains show enhanced sensitivity to the stray light contamination: e.g., $\lambda < 290$ nm in UV1, $\lambda < 320$ nm in UV2, both spectral ranges affected by the strong $O_3$ absorption, thus resulting in relatively low radiance readings and, as a consequence, a high sensitivity to the additive stray light component. Such sensitivity is augmented by the distinct wavelength dependence of the stray light contamination that may not be completely captured by the implemented correction algorithm: e.g., UV1 irradiances – see below. Based on outcome of the pre-flight tests, the stray light contamination is modeled in the L1B processor assuming a smooth (low-order polynomial) behavior in the spatial and spectral dimensions. The spatial stray light is measured at the dedicated stray light rows right below or above the imaging area of each CCD detector: the USA and LSA CCD areas in Figure 4. The signals from these rows are linearly interpolated over the entire CCD image. The spectral stray light dependence is evaluated at specific CCD columns, then interpolated at all wavelengths of a channel (UV1, UV2 or VIS), and whenever applicable extrapolated to the wavelengths of another channel. The spatial (row-wise) and spectral (column-wise) stray light components are combined to form a complete stray light envelope, which is eventually subtracted from the image. If the stray light signal is too large (i.e, the corrected radiances turn negative), flags are raised for the corresponding parts of the image. The results of such stray light flagging are shown in Figure 19. For the UV1 channel, the one with the highest stray light warning level (low radiances in the ozone-



absorbing domain), there is no significant increase in warnings over the mission time. Considering the potential influence of the Row Anomaly (RA; see Section 5 for more details) on the stray light estimates, one may conclude that the currently implemented straylight correction algorithm adequately accommodates such changes in UV1. On the other hand, despite the relatively low level of the flagged events, VIS channel shows some RA sensitivity: note the rapid increase in flagging at the

beginning of 2009 coincident with a major RA event.

Since the described procedure of stray light removal uses pre-flight characterization along with some general assumptions about the spatial and spectral stray light behavior, there is a need in independent estimates of the stray light contamination. This could be done relying on the sun as a relatively stable and predictable light source. In order to follow changes in the wavelength registration, as well as spectral line-profile shapes, we select multiple well-developed, relatively deep absorption

features (usually, blends of the solar absorption lines) spanning the UV1, UV2 and VIS ranges. In the daily irradiances and in every earthshine spectrum we measure: wavelength centroids of the absorption lines, full-widths at half-maxima (FWHM) and line depths. The line depth and FWHM are related to the radiances coming from the fixed-wavelength, relatively line-free spectral regions in the immediate vicinity of the measured absorption.

Line-profile estimates from the daily irradiance measurements are checked for $\pm 2\sigma$ outliers and binned into 3-month

averages. In the Earthshine data, at each orbit and each FOV the measurements are checked for $\pm 2\sigma$ outliers and averaged within 100-exposure orbital blocks. These orbital blocks are assembled into daily means. For each spectral line, each FOV and for a given range of solar zenith angles (SZA; the SZA bins are defined with 10-degree increments), the relatively RA-free measurements between 2004 – mid-2008 are used to estimate bi-annual seasonal variability patterns and subtract them from the individual measurements. The de-trended values are checked for outliers and binned into 3-month averages.

To produce the trends shown in Figure 20, we combine all the UV2 and VIS rows, and rows 1-13 from UV1 (reasons for the latter are discussed below), bin the values over 3 consecutive months, then normalize the line depth by the average line-depths values observed during the latest Solar minimum (March 2007 – August 2009). Both the line depths and, to a lesser extent, FWHMs (not shown) of the absorption blends follow the predictable changes related to the Solar Cycle (see more details in Marchenko and Deland (2014)). In essence, practically all absorption lines in the OMI irradiances are getting

progressively shallower with the gradually (years) increasing solar activity levels. This creates the inverted-U shapes seen in Figure 20. The long-term changes are far more pronounced in the UV1 range compared to VIS, in line with the expected Solar Cycle behavior. If there are any instrumental trends, then at this point they cannot be clearly disentangled from the anticipated solar-related changes in the UV2 and VIS ranges. The relative changes (i.e., the deviations from the expected inverted-U shape) in the UV1 line depths point to possible ~0.5%-1.0% instrumental trends, especially considering the

temporal behavior of the 302.12 nm blend, to be compared to the UV2 lines closely following the expected transformations. We also performed (not shown here) line-depth measurements for various spectral features in the UV2 range of the Earthshine spectra. In general, the earthshine trends conform to the inverted-U shapes seen in the UV2 irradiances (the middle panel in Figure 20). I.e., in radiances the gradual line-depth changes are also mostly driven by the long-term (years) Solar variability. However, we noticed some subtle deviations from the expected trends, most likely related to gradual



straylight changes not properly accounted for by the currently adopted (Collection 3) approach. Considering the measured line-depth values, as well as magnitudes of the deviations, we assume that such deviations may be caused by ~0.5% long-term changes in the UV2 straylight levels, in line with the relative ~0.5%-1% difference in the long-term trends derived from the low-reflectivity and high-reflectivity subsets of the UV2 radiances (see below).

## 4.4. Instrument temperatures

The temperature of the optical bench impacts the wavelength registration. The design of the optical bench is such that thermal fluctuations of the optical bench should have a minimal effect on wavelength registration. In the OMI case there is a small, however detectable relation between the two quantities. The small seasonal variability in the trend in wavelength registration (see section 6.3) can be directly related to temperature fluctuations. Besides, the dark-current readings depend on temperature. In general, when the temperature of the detector rises by 10 degrees, the average dark current doubles. This calls for low and very stable operational temperatures of a CCD detector.

The OMI optical bench (OPB) is cooled by a passive radiator plate. Without additional heating the temperature of the OPB would be about 255 K. Passive heaters warm the OPB to the operational temperature of 264 K. The temperature of the CCDs is controlled with active heaters in a closed-loop feedback system. The operational temperature is 265 K, controllable to +/- 10 mK. Figure 21 shows the trend in the temperature of the OPB and the UV CCD detector. The trend in the temperature of the VIS channel looks similar to that of the UV channel, with a steady increase of the temperatures early in the mission, which flattens after 2010. The trend in CCD temperature very closely follows that of the OPB, albeit the absolute change is far smaller (note the different scales for CCD and OPB temperatures). The temperature of the OPB can be controlled with a 1 degree tolerance limit. The temperature setting was not changed during the mission. Therefore the 1 degree increase early in the mission is large and should have triggered system's response. Consecutively, the temperature of the CCD has increased by 40 mK, thus far exceeding the tolerances of the controlling system. The lack of timely adjustments in both controlling systems remains unexplained. This, however, does not impact the OMI performance in any major way (see below).

## 4.5. Voltages

The Electronics Unit (ELU) monitors a number of internal voltage values. Analysis of the voltage data has shown that fluctuations over the mission are small. The largest fluctuation that was seen in the 5 and 12 volt lines was 0.07 %. There are two voltage parameters that show larger variability. The WLS voltage shows changes up to 0.5 %, and the Test voltage changes by 12%. Since the Test voltage is not used for nominal operations, this increase poses no problem for radiances, irradiances and Level 2 product retrieval.



## 5.  Anomalous behavior

Since June 2007 (the currently accepted date; there are some, though very limited, indications of an even earlier onset of the anomaly) OMI has suffered from the so-called "Row Anomaly" (RA) phenomenon. In this anomaly certain Earth-observing cross-track FOVs (rows) are seemingly blocked, resulting in abnormally low radiance readings. The most probable cause of

blocking is a partial external obscuration of the radiance port by a piece of loose Multi-Layer Insulation (MLI) of the instrument itself, but this is not certain. The first signs of the anomaly were detected in rows 54 and 55 (1-based). These rows remain affected ever since. Since May 2008 the anomaly affects image rows 38-42 (see Figure 22). At the time of writing the anomaly was relatively stable, permanently affecting UV2 rows 25-42 and 54-55, and occasionally spreading to rows 43-53. Figure 22 depicts the RA evolution in the UV1 and UV2 channels. The VIS behavior is somewhat similar as

UV2, however showing different degrees of involvement for the rows in the immediate vicinity to the main RA domain (defined as rows 25-42). The row anomaly affects the data in four different ways :

- *Blockage effect*

    Several rows (cross track viewing angles) have a decrease in signal strength. This decrease is assumed to be caused by something blocking the nadir port of OMI. The blockage effect is a multiplicative, wavelength-dependent factor.

- *Solar light contamination*

    Several rows show increased signal level. This increase predominantly happens in the northern part of the orbit, in apparent relevance to the incident sunlight. It is assumed that something outside the nadir port is reflecting sunlight into the instrument. This could be a piece of loose MLI. This increase in signal level has an additive, wavelength-dependent effect.

- *Wavelength shift*

    The partial blocking of the nadir port results in inhomogeneous illumination of the OMI spectral slit. This causes a slight change in the instrument spectral response function, changing wavelength registration.

- *Earth radiance from outside nominal field of view*

    Several rows may show increased signal levels at certain parts of orbit. This is caused by the earthshine from outside of

the nominal FOV reflected into the nadir port. This is an additive factor with time/FOV-dependent terms, thus the most elusive row anomaly effect.

Based on level 1 data, a daily automatic analysis distinguishes between these four row anomaly effects. A warning flagging scheme is based on the multitude of parameters provided by such analysis. The most influential contributing factors are: number of negative reflectances, number of overly large reflectances, the reflectance histogram, the mean scaled wavelength

shift, and wavelength fit failure count. This flagging scheme is added to the level 1B product. If the daily analysis results shows significant short-term changes, the flagging scheme is adapted manually. The number of the affected rows has increased since the first appearance of the row anomaly in 2007. Figure 22 shows the affected UV1 and UV2 rows. The VIS channel looks similar to the UV2 channel.



The RA effects grow progressively larger, with pronounced seasonal modulation, in the northern parts of the OMI orbits , when sunlight is coupled into the instrument via the radiance port. Table 3 shows the percentage of rows that is affected for all orbit phases, and for the northern parts of orbits, with a noticeable 100% involvement of the UV1 channel. The row anomaly effect is not corrected by the L1B processor. The RA flags are included in the affected L1B.

We also performed an independent analysis of the OMI radiances, applying the following procedure. Typical OMI orbit provides ~1640 2-second exposures. These are binned into 50-exposure blocks. This initial binning assures better S/N for the spectra obtained over areas with low surface reflectivities and/or high solar zenith angles. Besides, the relatively small size of the bin keeps the gradual orbital drifts in the wavelength registration (thermal flexure) well below ~0.005 nm for exposures within the bin. For the UV2 data, observed scenes are partitioned into 3 reflectivity categories: r < 15%, 15% ≤ r ≤

60%, r > 60%. Such partitioning provides roughly (by a factor of few) comparable sample sizes. The scene reflectivities at λ=331 and 360 nm are estimated by the OMTO3 (total ozone) algorithm. The 50-exposure averages for each reflectivity category are corrected (normalized) for changes related to variable solar zenith angles, then interpolated to a common wavelength grid. The data in each 50-exposure block are wavelength-binned (~0.5 nm bins) around the relatively line-free spectral areas, then assembled into multiple-day averages (currently, in 15 day blocks; for illustration purposes, 90-day

averages are shown in some plots). For each FOV (row), each orbital 50-exposure block and each binned wavelength we derive the bi-annual 'climatology' based on the data from the row anomaly-free epoch between 2004 and ~mid 2008. At the last step we remove (subtract) these periodic patterns and bin the de-trended values into broad 30-degree latitudinal zones. At every binning step the data are checked for ±2σ outliers. The procedure slightly differs for the UV1 and VIS radiances. The UV1 data are not partitioned into different reflectivity groups, while the VIS radiances are segregated into the low-,

mid- and high-reflectivity (at λ=388 nm) categories at r<10% and r>70% thresholds. To augment S/N (here the noise may be related to the inherently low signal level at the UV1 wavelengths; the 'noise' could also be produced by the variable scenery in the VIS range), we use much broader, 2-3 nm, wavelength bins in UV1 and VIS. We do not account for any SZA-related variability in the VIS data, delaying the removal of relevant trends until the 'climatology' subtraction step (as above).

   In Figure 23 we plot the de-trended, binned UV1 and UV2 radiances for selected rows. The shown rows are very close to the 

main row-anomaly area (e.g., mainly image rows 25-42, 54 and 55 in UV2, with occasional broadening of the row anomaly (RA)-affected area towards image rows 43-52). These 'bordering' rows demonstrate relatively weak reaction to the on-going RA. Note the dominance of the blocking in the southern-hemisphere domain, lat=-45 deg, and the interplay of the blocking (line-of-sight obscuration) and the solar straylight (December-January spikes at lat=45) in the northern-hemisphere radiances. The solar straylight is far more pronounced in the UV1 range, causing saturation of the UV1 detector at some

rows, as well as affecting practically all UV1 rows at northern latitudes. E.g., UV1 rows 12-22 (counting the row numbers from 1) are saturated al all wavelengths at high northern latitudes, lat >=10N. The rows 23-27 show diminishing (with the increasing row number and increasing UV1 wavelengths) swathes of saturated radiances. Rows 1-11 and 28-30 can be considered as practically saturation-free in the λ > 295 nm domain.





Figure 24 shows the de-trended row-, altitude- and wavelength-binned VIS radiances for the low-reflectance (r < 10% at λ=388 nm) sub-sample of the data. Besides the remarkably low instrument degradation, ~1-1.5% between 2004 and 2015, one may notice ~1-3% dip around 2009-2010. The feature becomes more pronounced as the row index moves closer to the RA-affected areas (rows > 23). This 2009-2010 dip is seemingly absent in the high-reflectance category (r > 70%; not

shown) with the latter being far less sensitive to the RA-related changes in the straylight level. Hence, at some epochs (as shown for VIS) or at some orbital phases (as happens in UV1 λ < 300 nm radiances obtained at SZA > 45 deg in the northern hemisphere) the RA-related changes may affect FOVs (rows) beyond the limits routinely flagged as RA-contaminated (as in Figure 22). Various OMI L2 products show different sensitivity to the RA phenomena, thus calling for a cautious interpretation of the OMI data in the 'bordering' areas, e.g., UV2 rows 24, 44-52.

A single reliable method for the detection of the row anomaly has proven difficult to establish because the effects of the anomaly on radiances are complex and each science algorithm has its own sensitivities to radiance error that are difficult to capture with a single detection technique. The KNMI methods for detection through analysis of the OMI L1B radiances directly worked well to flag bad data from their L2 products, but did not satisfactorily remove affected retrievals in some NASA L2 products. Therefore an additional method was developed to determine the affected rows for the NASA algorithms,

which is based on analysis of errors detected in the NASA TOMS L2 total ozone product. The NASA team developed its own row anomaly detection scheme that identifies instrument error using a statistical analysis of total column ozone error. Total ozone anomalies are detected using data averaged in 5 degree zonal mean bands by comparing the row-to-row behavior of the data to a baseline OMI dataset similarly constructed from data collected prior to the onset of the row anomaly (Haffner, 2012). Total ozone is a good basis for the anomaly detection because algorithm errors are relatively well

understood and the mean geophysical behavior of total ozone can be thoroughly characterized. Though the NASA flags were originally designed for total ozone, they also work well for other products such as $SO_2$, which is very sensitive to radiance errors, and also the OMI aerosol optical depth and single scattering albedo product derived from the VIS detector. A comparison between the results from the NASA and KNMI flagging results is shown in Figure 25. In this figure, no distinction is made between the four different row anomaly effects or if a row is flagged for the entire orbit or just a part of

the orbit. This figure nicely summarizes the percentage of OMI detector rows that are affected by the anomaly as a function of time, as detected in the KNMI L1B monitoring approach and the NASA method. Both indicators track the anomaly similarly, but differences do exist in how the flags are set for some data.

## 6.  Long-term calibration

### 6.1.  Status of current Collection 3 L1B products

The L1B products that are produced are part of the Collection 3 data. Collection 3 data has started on February 1, 2010 with the introduction of version 1.1.3 of the L1B processing software (GDPS) (Dobber et al, 2008a). When version 1.1.3 was





introduced, all data since the beginning of the OMI mission has been reprocessed with this version. The main improvements in version 3 are: a more elaborate flagging of the row anomaly effects, new wavelength fit coefficients, improved stray light correction in UV2, and an improved noise calculation. A one-time adjustment to the radiometric calibration was also applied. There were no changes in the basic flow of corrections on the data products. These corrections are shown in Table 4. More

extensive information with e.g. flagging functions can be found in Oord et al (2006). The generic functions in the table are executed for all measurements. Depending on the measurement type an extra series of correction functions is applied. There has been an effort to make correction functions for the row anomaly effects, but these corrections did not give the desired results. It is difficult to separate the different row anomaly effects, and therefore they could not be corrected satisfactorily. There are only flagging functions for the row anomaly effects.

**6.2.    Radiometric calibration**

This section addresses changes in instrument radiometric calibration as observed in the solar measurements and the Earth radiance measurements. Each observational port may provide an independent view of sensor changes since launch that may not be necessarily consistent. This is particularly true for OMI, where the optical paths differ for Earth-view and solar measurements. The challenge is to reconcile these differences and to describe as accurately as possible the calibration

changes in the Earth radiance path. We begin with a discussion of the solar measurements.

Solar calibration measurements are performed every day. In a solar calibration measurement the sunlight passes via the mesh through the opened solar aperture onto a reflective diffuser: either the aluminized fused silica (QVD hereafter), or two pure Aluminum diffusers. The reflected sunlight is coupled into the instrument telescope via the Folding Mirror (see Figure 5).

The relative solar signal in the UV1 channel for the three diffusers is shown in Figure 26. In this figure a solar measurement

is divided by a reference solar measurement from the beginning of the mission. The average of this ratio for the UV1 channel is calculated and shown in the figure. If we assume the three react similarly to solar exposure, their differences appear to be related to their frequency of exposure. The QVD is used every day, the regular Al diffuser once per week, and the backup Al diffuser once per month. This leads to less degradation of the aluminum diffusers. The changes observed in all channels are provided in Table 5. For the UV1 channel, the signal change is 6 % for QVD, 3 % for regular Al and 2.5 % for backup Al.

These are overall signal changes of the complete instrument. Since the backup diffuser is used so little, the signal change of 2.5 % can be attributed to the complete instrument.

To further substantiate our hypothesis of QVD optical degradation we zoom in into individual wavelength bands in Figure 27. The bands with the shortest wavelengths have the largest degradation. An exception to this is the 372-376 nm band in UV2 (not shown), which tends to degrade slightly (by ~0.5% over 10 years) faster than expected for the particular

wavelength range.

We attribute the accelerated degradation of Row 20 in UV1 starting 2009 to scattered sunlight during northern hemisphere Earth-view measurements. This is described as the solar contamination effect in Section 5. Assuming this is the cause, the



change is likely occurring in the telescope assembly. This follows because the diffusers, as well as the folding mirror are not in the optical path during Earth-view measurements (Figure 5), hence they are not exposed to the anomalously scattered solar light. The primary telescope mirror is bypassed for the solar measurements, so the Row 20 anomaly in Figure 27 is likely caused by accelerated degradation of the secondary telescope mirror. We surmise that the degradation of the primary

mirror is even greater for the RA-affected across-track positions since it is the first optical element exposed to the RA-scattered solar light. The interference with other row anomaly effects makes it impossible to verify this hypothesis.

We can isolate the optical degradation of the solar diffusers by comparing the signal changes observed with each. Figure 28 shows the fractional change in the QVD per hour of solar exposure relative to the other two diffusers. The close agreement between QVD changes derived from the Regular and Backup diffusers is an indication that neither has degraded

significantly. If we assume the Regular and Backup have similar degradation rates the former should be degraded more than the latter in a ratio of 32/7. A substantial degradation of either would result in a separation of the blue and green points in Figure 28.

Closer examination of Regular diffuser change relative to the Backup reveals a rate of $1.10^{-4}$ hour$^{-1}$ at 265 nm, which is significantly less than the $3.10^{-4}$ hour$^{-1}$ observed for Volume diffuser change. This difference is important because the multi-

15 diffuser approach to calibration relies on equal degradation rates. For the results presented here we have assumed the Backup diffuser has not degraded, but in fact a small correction is required for its change based on the Regular diffuser degradation rate. Since that rate is already very low it is unlikely that an error in the Backup degradation rate, even if it were half that of the Regular's, would result in a significant calibration error. Still, these results should serve as a warning to minimize the exposure of the least-used diffuser lest uncertainties in its degradation rate become a significant component in the calibration

error budget.

The observed QVD degradation rates are similar to those seen for several of the SBUV2 instruments, though the OMI QVD appears to have a steeper wavelength dependence. It is noteworthy that the OMI solar measurements employ a protective mesh in front of the diffuser that attenuates the incident solar irradiance by approximately a factor of 10. This implies that the QVD degradation rate per equivalent solar exposure is much larger than that of the SBUV2 diffusers and larger even than

25 that of the TOMS diffusers (see Jaross et al (1998)). The design and operational factors affecting degradation rates are complex and to this day not fully understood. This underscores the importance of maintaining low exposure frequencies until on-orbit rates become clear.

The degradation of the OMI instrument downstream of the diffusers is approximated by the change observed in the Backup measurements (see Table 5) because the expected degradation of that diffuser is so small. To assess how these non-diffuser

changes affect the OMI Earth-view measurements requires an independent estimate of measured radiance change. The estimated changes, shown in Figure 29, are assessed by removing common seasonal and other cyclic variations. No attempt has been made to remove any putative long-term geophysical changes from the radiances.

Instrument changes related to Earth-view measurements are summarized in Figure 30. Signal changes for the Solar QVD measurements, shown in the same plot, are significantly greater. Note that in the UV region ($\lambda < 300$ nm) natural variations





can exceed 1% at typical Solar Cycle time scales (>~5 years: Marchenko and Deland (2014)). The shown QVD degradation changes are corrected for this variability. Falling substantially below the QVD values, the Earth radiance rates are nevertheless consistent with solar signal changes measured with the Backup diffuser(cf. Table 5). This suggests that for rows not affected by the Row Anomaly the change in the primary telescope mirror is negligible, and the change in radiometric calibration can be accurately estimated using the Backup diffuser solar measurements, once the Solar variability factors are taken into consideration.

The derived Earth radiance changes are confirmed by observing measured signals (Figure 31) over Greenland and Antarctica. Assuming that the mean reflectivity of the ice surfaces has not changed over the OMI mission, we conclude that the optics and detector have changed by ~1-1.5% at 360 nm in UV2 for rows far from the row anomaly blockage. Substantial changes are observed near the RA-affected nadir view (right panel in Figure 31), where the blockage is the greatest.

This conclusion is substantiated by the long-term trends seen in the OMI radiances (Figure 29). In this figure we show de-trended and wavelength-, latitude-, time- and row-binned radiances. As an example, for UV1 we select all the available data for image rows=6-10 which are practically unaffected by RA at southern-hemisphere latitudes. We show image row=2-7 UV2 trends for all latitudes and the radiances from the low-reflectivity (<10%) subsample of the data. In addition, we show the trend for the high-reflectivity (>80%) group for the 357-373 nm wavelength bin, keeping in mind that the onset of a major RA event in January 2009 may have changed the OMI straylight levels due to the physical blocking of some Earth-viewing angles, as well as additional scattering of the Sun- and Earth-light. Hence, if the currently implemented straylight correction does not adequately capture the RA-related changes, one should see different temporal behavior among the high- and low-reflectance sub-samples, with the latter on average ~5 times more susceptible for the straylight effect than the former. Indeed, we see close agreement between the high-reflectivity (dotted red line in Figure 29) and low-reflectivity (full red line) UV2 trends which abruptly changes at the beginning of 2009. However, the overall effect is rather small, reaching ~1% for the long-wavelength UV2 range and consistently staying below 1% (our sensitivity limit in detecting the long-term trends in radiances) in the 310-350 nm UV2 range. Chromatic terms in the radiance trends are relatively small for UV1 and UV2 (<~2% and <~1% over the mission time, respectively) and practically absent (<0.5%, our sensitivity limit in detecting the chromatic trends) in VIS.

We directly examined the ratios of sun-normalized radiances measured at wavelengths separated by several nanometers to confirm there is little change in the spectral dependence of OMI's overall radiance calibration over the course of the mission. The data shown in Figure 32, for three of the OMI detector rows unaffected by the row anomaly, are selected to minimize natural sources of trend and variability in the ratios of radiances. The main geophysical effects which introduce time-varying spectral dependence in back-scattered Earth radiances are Rayleigh and aerosol scattering, trace gas absorption, and Raman scattering. Because these effects are much larger than the changes in the instrument spectral dependence, and they are highly variable over space and time, particularly over shorter scales, it is useful to isolate certain data so the impact of these effects on the radiances is reduced. 354 nm radiances are unaffected by trace gas absorption, and while 340 and 380 nm radiances have minor $O_3$ and $O_2$-$O_2$ absorption, these effects are small and we limit their impacts on radiance trends by restricting our





data to the tropics where the variability in total ozone is small. We select data measured over the central Pacific in the tropics, a region removed from sources of dust and smoke that would contaminate the spectral ratios. To reduce the effect of Rayleigh and Raman scattering we restrict ourselves to data measured over very bright convective clouds, that we select by requiring that the top of the atmosphere (TOA) reflectance, $\rho$ at 340 nm is greater than 0.9, and we calculate $\rho = I/F \cdot$

$\pi \sec \theta_0$ where $\theta_0$ is the solar zenith angle and radiance $I$ is normalized by solar flux $F$. The large values of TOA reflectance over the tropical oceans are associated with mature deep convective clouds whose cloud top pressures are near 200 hPa (~12 km) and optical centroid (effective) cloud pressures inside the cloud are near 500 hPa (5.6 km) because of appreciable penetration into the cloud of photons subsequently scattered back to the top of the atmosphere (Vasilkov et al., 2008 and Ahmad et al., 2004). These clouds are of such high reflectance and at altitudes well above a significant portion of the

atmosphere that the ratio of direct to diffuse scattered radiation received by OMI is significantly enhanced by the presence of these clouds, which reduces the variation in Rayleigh scattering due to changes in solar zenith angle with season. The largest values of solar zenith angle for the data considered here are 50°.

The results in Figure 32 show the monthly means of the wavelength ratios of 340 to 354 nm on the UV-2 detector, and 354 to 380 nm from the VIS detector, for three detector rows over the course of the OMI mission (useful 354 nm measurements

are made on both the UV-2 and VIS detectors). The trends in these ratios are less than 0.5% per decade.

Small seasonal and interannual variations remain in the radiance ratios despite our efforts to minimize the wavelength dependent geophysical effects on the variability of these data. These variations are most from remaining geophysical effects such as residual solar zenith angle dependence, variation in cloudiness, and possibly aerosol contamination caused by volcanic events. They are not thought to be related to detector performance. The somewhat greater amplitude of the

variations seen for the VIS detector ratios can be explained by the fact that the leading spectrally dependent effects at these wavelengths increase with greater wavelength separation, and the wavelengths on VIS are spaced by 26 nm whereas those on UV-2 are separated by 14 nm. This analysis is limited to the UV-2 and VIS detectors because radiances measured by UV-1 are much more sensitive to ozone, and ozone absorption cross-section varies dramatically over just a few nanometers in that detector's spectral range.

**6.3. Spectral Calibration**

During the mission spectral calibration is performed as described in section 3.2 and Appendix A. The in-flight spectral calibration coefficients are calculated using two different methods. In the first method the pre-flight calibration parameters are modified by a function using the optical bench temperature, and a correction for inhomogeneous slit illumination. For the second method the calibration coefficients are determined using a fit of the measured spectrum with a high-resolution solar

spectrum, augmented with an ozone spectrum and a Ring spectrum.

The changes of calibration parameters based on the fit results (the second wavelength calibration method) are shown in Figure 33. Here we plot the first polynomial coefficient ($c_{fit,0}(j)$ in equation A.16) for image row j=15 for UV1, and image



row j=30 for UV2 and VIS. Note that the radiance measurements at these rows are affected by the row anomaly, which greatly exacerbates the scatter in the UV1 values after the major January 2009 RA event. This particular polynomial coefficient (in the plots we show the daily averages) is used to demonstrate the long-term stability of the instrument. The wavelength registration gradually drifts by 0.015 nm in UV1, and for the UV2 and VIS channel it shows remarkable

stability, changing by ~0.002 nm (~1/100 of a pixel) over the 10-year span. We also see that the trend in the shift of the wavelength registration follows that of the temperature of the optical bench (cf. Figure 21). Therefore the initial expectation that the wavelength registration depends mainly on the temperature of the optical bench, seems to be fulfilled.

We complement the findings from Figure 33 by trending the wavelength registration provided via the first calibration method, i.e., the pre-flight calibration adjusted by the OPB temperature and, whenever applicable, the inhomogeneous slit

illumination. As an example, we take the daily OMI irradiance measurements and for each row select prominent spectral lines spanning the sensitivity ranges of the OMI spectral channels. For each row in each channel we calculate centroids of these prominent absorptions. For a given line, in all rows the calculated centroids show similar, to within sensitivity limits, time dependencies. Hence, we average all the UV2 and VIS rows, however limiting the averages to the UV1 image rows 1-13, thus avoiding the FOVs experiencing anomalous degradation rates (see Figure 27). We additionally average values over

3 consecutive months and subtract the early-mission estimates from the line-centroid measurements. Figure 34 shows changes in the line centroids for the selected representative lines. Over the mission time, line positions gradually shift at <~0.005 nm pace in UV1, save the ~290-300 nm region where the drift practically triples. The UV2 and VIS channels show progressively lower changes, with long-term trends in the latter not exceeding the scatter levels, ±0.001 nm, probably related to the yearly oscillations of the OPB temperature.

Long-term (mission time) and short-term (orbital) stability of the instrument spectral response function is deemed important for reliable, unbiased retrievals of the atmospheric trace-gas properties. Changes in the instrument spectral response affect depths and widths of the detected spectral features. In Figure 35 we show variations of the line-profile parameters derived from radiances for the line blend around $\lambda$=336.1 nm and the UV2 row#5. Each panel shows the differences between the latitude- and time-binned early-orbit (lat=-60 to -50) and late-orbit (lat=40 to 50) line-profile parameters. The orbit-

differentiated wavelength registration and FWHMs go through relatively minor (±0.001 nm) seasonal changes which we deem negligible in comparison to the 0.14 nm UV2 sampling rate. The line-depth variability show clear ±0.2% seasonal fluctuations, most likely related to changes in the Ring line-filling factors, with their direct proportionality to the seasonably changing (Solar elevation for a given latitude) atmospheric path-lengths. The line centroids are also involved in ±0.001 nm seasonal cycling. However, such fluctuations should be regarded as negligible in comparison to the 0.142 nm sampling rate

in the UV2 spectra. Hence, we may conclude that instrumental factors do not introduce observable (i.e., exceeding our sensitivity limits) spectral response changes along the OMI orbit. Nor such factors cause any long-term (mission time) instrumental trends exceeding ~0.2% in measurements of the UV2 and VIS absorption features (see Figure 20).



## 7. Conclusions, summary and outlook

Analyzing long-term trends in the OMI L1B products, as well as values of calibration parameters, we conclude that, apart from the ongoing Row Anomaly, the instrument continues to perform well. Though still gradually unraveling, the Row Anomaly remains relatively stable since the latest incident in the summer-fall 2011.

One of the most noticeable trends is the sevenfold dark current increase. The RTS warnings, which are closely related to the dark current readings, have increased to 0.7 % of the CCD pixels used for data acquisition, or to 5.6% if one to consider the binning factors (8 CCD pixels = 1 row) used in the L1 and L2 OMI products. Bad pixel warning has increased to 1.4 % (11.2%). The output of the monitoring lamps, LED and WLS, decreased to 85 % and 90 % of the early-mission values, respectively. However, this has not hampered the routine calibration activities. There are some fluctuations in the gain ratios

and electronic offsets, but they are small and do not impact the signal processing routines. The nonlinearity warnings and straylight warnings remain stable.

The gradual decrease in the irradiance values can be primarily attributed to degradation in the solar diffusers. The daily measured irradiances from the QVD diffuser decreased by 6 % in the UV1 channel. The output from the Al diffuser, which is used once per week, diminished by 3 % in the same wavelength range. The rarely (once per month) used backup Al

diffuser has degraded by 2.5 % in the UV1 channel. Since the exposure times for the weekly and monthly calibration cycles differ by 4 times, it can be safely assumed that the registered 2.5 % degradation of the backup Al diffuser is attributable to the complete optical pathway of the OMI irradiance measurements (Figure 2). We also register the substantially (~twice) higher degradation pace in the UV1 rows affected by the row anomaly. The longer-wavelength OMI channels show progressively lower degradation, with daily QVD values dropping by ~3% over the mission time.

In the RA-free areas the trends in the OMI radiances point to surprisingly small changes, with good consistency among the UV1, UV2 and VIS channels: the UV region shows a ~2% downward trend, while at visible wavelength the long-term changes amount to 1-1.5% over the mission time.

The long-term wavelength drifts in the UV2 and VIS channels do not exceed 0.005 nm, attesting to excellent thermal/mechanical stability of the instrument. Gradual drifts in the UV1 range amount to 0.015 nm, with some evidence of

25 wavelength dependence.

We perceive the Row Anomaly as the most formidable instrument problem that renders unusable a significant proportion of the RA-affected rows (FOVs). The anomaly was unequivocally detected in two rows in June 2007. Alternative approaches point to possibility of an even earlier incursion around the fall 2005 – winter 2006. In May 2008 a large new group of rows became affected. The row anomaly continued to develop since then, with the particularly swift changes around January 2009

and the early fall of 2011. It is relatively stable since then. The latest small increase in the affected rows dates from August 2014. Overall, the numbers of the RA-affected rows depend on the OMI channel, with radically different latitudinal and seasonal behavior in the UV1 and UV2 channels, and comparable patterns in the UV2 and VIS ranges. Considering complexity of the temporal and spatial changes, in the summary Table 6 we provide only indicative estimates of the RA-



affected rows. E.g., currently, about 33 % of the UV2 rows are affected in the southern-equatorial parts of the OMI orbit. This increases to ~57% in the northern hemisphere. These estimates are comparable to the VIS numbers, though very different in the UV1 case, where all rows are affected at the middle-to-high northern latitudes. Users of the OMI data are advised to discard the affected rows in accordance with the warning flags provided in the L1B products, though one may
notice a broad range of RA sensitivity among the different L2 OMI products.

As appears form the presented data, OMI ages very gracefully, showing remarkably low radiance degradation and high wavelength stability. The most serious concern is the developing Row Anomaly. However, its spread has stabilized since the last rapid development in the fall of 2011. Assuming *status quo*, one may expect that the instrument can deliver useful science data for 5-10 additional years.

**Acknowledgements**

The authors would like to acknowledge the work of the large team of colleagues at KNMI, FMI, NASA, and in industry, that have contributed to the success of the OMI instrument. This work could not have been done without funding from the Netherlands Space Organization and NASA.

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



## Appendix A: Wavelength registration approach

This appendix describes how the wavelength registration of OMI has been performed during the onground calibration period, and how it is executed in flight. During the onground calibration period the wavelength calibration has been performed using a PtCrNeAr spectral line source. In the analysis of the calibration data small wavelength windows were used that contained exactly one spectral line, of which the wavelength is accurately known. In the UV1 channel 5 windows were used, and both in the UV2 and VIS channels 9 windows. In such a window the spectral calibration was performed by fitting the measured spectrum to a Gaussian function in the UV1 channel, and a sum of a Gaussian and a function with a flatter top in the UV2 and VIS channels:

$$F(x) = e^{-2\left(\frac{x-x_0}{W}\right)^2} + e^{-2\left(\frac{x-x_0}{W}\right)^4}$$

(A.1)

The fits were performed in pixel space. In this way for all windows the peak position of the spectral line in pixel space was determined. The peak positions together with their wavelengths are fitted to a fourth order polynomial to obtain the wavelength scale. This wavelength scale is used to calculate the wavelength of a given column number in the L1B processor. During the mission the wavelength assignment is done using two methods. Both for the radiance and irradiance measurements the wavelength coefficients are obtained. In the first method the wavelength assignment is based on pre-launch and early in-orbit wavelength calibration parameters. These parameters are modified as a function of Optical Bench temperature:

$$c_{2,n}(j) = c_n(j) + d_n(j)\left(T_{OPB} - T_{sc\_ref}\right) + e_n(j)\left(T_{OPB} - T_{sc\_ref}\right)^2$$

(A.2)

where $c_n(j)$ are the pre-launch wavelength calibration parameters, $T_{OPB}$ is the temperature of the optical bench during the measurement, $T_{sc\_ref}$ is a reference temperature, $d_n(j)$ and $e_n(j)$ are wavelength temperature coefficients and j is row number. In the UV2 and VIS channel this function is corrected for wavelength shifts that result from inhomogeneous slit illumination. This correction makes use of small pixel data. Normal radiance and irradiance data is data that has been binned on the CCD and coadded in the Electronics Unit. Both operations result in increased signal to noise ratio at the expense of spatial and temporal resolution. It also helps in reducing the datarate to ground. A typical binning factor is 8, and a typical coadding factor is 5. Small pixel data is data that has not been coadded. The correction function for inhomogeneous slit illumination is as follows:

$$Q(j) = 2.\frac{S_{SPC}(N_{co-addition} - 1, j) - S_{SPC}(0, j)}{S_{SPC}(N_{co-addition} - 1, j) + S_{SPC}(0, j)}$$

(A.3)

$$c_{2,n}(j) = c_{2,n}(j) + b_{OPF}(j).Q(j)$$

(A.4)



Where $S_{SPC}$ is a matrix containing small pixel data, $N_{co\text{-}addition}$ is the number of coadditions used for this measurement, $b_{OPF}(j)$ are transformation parameters and j is row number. With the wavelength coefficients $c_{2,n}(j)$ a wavelength map is calculated using:

$$\lambda(i,j) = \sum_{n=0}^{N} c_{2,n}(j)\left(i - ii_{ref}\right)^n$$

(A.5)

where i is column number, j is row number, N is number of polynomial coefficients (typically 4) and $ii_{ref}$ is a reference column for the spectral calibration polynomial coefficients. The wavelength coefficients $c_{2,n}(j)$ and the reference column are stored in the L1B radiance and irradiance output products.

For the second method, the wavelength calibration is performed by fitting an accurately known solar spectrum, an ozone absorption spectrum and a Ring spectrum to the measured spectrum. The reference spectrum is divided in 8 windows for the UV1 channel, 18 windows for the UV2 channel and 22 windows for the VIS channel. These windows contain characteristic features of the solar spectrum like Fraunhofer lines. For each window a fit of the measured spectrum with the reference spectrum is performed in wavelength space. The boundaries of a window are given by $\lambda_{min}$ and $\lambda_{max}$. An initial estimate of the wavelength scale is given by $\lambda(i,j)$ where i and j are column and row index of the CCD detector. The result of the fit process will be a wavelength scale λ' where

$$\lambda'\left(\lambda(i,j)\right) = \lambda(i,j) + \lambda_{shift} + (\lambda(i,j) - \lambda_{center}).\lambda_{squeeze}$$

(A.6)

and

$$\lambda_{center} = \frac{\lambda_{min} + \lambda_{max}}{2}$$

(A.7)

$\lambda_{shift}$ and $\lambda_{squeeze}$ are non-linear fit parameters. The fit process is executed by minimizing the spectral calibration merit function

$$\chi^2_{window} = \sum_{\lambda=\lambda_{min}}^{\lambda_{max}} \left(\frac{S_{measurement}(\lambda) - S_{fit}(\lambda')}{\sigma(\lambda)}\right)^2$$

(A.8)

where

$$S_{measurement}(\lambda) = {}^e\log S\left(\lambda(i,j)\right)$$

(A.9)

and




$$S\big(\lambda(i,j)\big) = S(i,j) \tag{A.10}$$

which is the signal at detector location (i,j), and

$$\sigma(\lambda) = \frac{Noise\big(\lambda(i,j)\big)}{S\big(\lambda(i,j)\big)} \tag{A.11}$$

The fit function is

$$S_{fit}(\lambda') = a_0 . sf_{DOAS,0} . {}^e log\big(I_{sun}(\lambda')\big) - a_1 . sf_{DOAS,1} . \sigma_{ozone}(\lambda') \tag{A.12}$$

$$-a_2 . sf_{DOAS,2} . Ring_{DOAS}(\lambda') + \sum_{n=3}^{5} a_n (\lambda' - \lambda_{center})^{n-3} sf_{DOAS,n}$$

where $I_{sun}(\lambda')$ is a high-resolution sun spectrum, $\sigma_{ozone}(\lambda')$ is an ozone absorption spectrum, $Ring_{DOAS}(\lambda')$ is a Ring spectrum, $a_n$ are the fit parameters and $sf_{DOAS,n}$ are scaling parameters. After minimalization of the merit function, a column / wavelength / precision triplet is calculated for each window w:

$$i_{win}(w) = \frac{i_{min}(w) + i_{max}(w)}{2} \tag{A.13}$$

Where $i_{min}(w)$ and $i_{max}(w)$ are the columns with a wavelength closest to $\lambda_{min}$ and $\lambda_{max}$ of the window,

$$\lambda_{win}(w) = \lambda'\big(\lambda(i_{win}(w),j)\big) \tag{A.14}$$

The precision $\sigma_{win}(w)$ is taken equal to the covariance of the result of the minimization of the merit function. The sets of triplets of all windows in a row are taken for a second polynomial fit. The polynomial is fitted by minimization of the following merit function:

$$\chi^2_{row} = \sum_{w=0}^{W} \left( \frac{\lambda_{win}(w) - \lambda_{fit}\big(i_{win}(w)\big)}{\sigma_{win}(w)} \right)^2 \tag{A.15}$$

with

$$\lambda_{fit}(i) = \sum_{p=0}^{P} c_{fit,p}(j)\big(i - ii_{ref}\big)^p \tag{A.16}$$

and $c_{fit,p}(j)$ are the polynomial coefficients for a row j, $ii_{ref}$ is a reference column for the spectral calibration polynomial coefficients, P is the number of polynomial coefficients and W is the number of windows. With these coefficients the wavelength of a pixel can be calculated for a given column / row index (i,j). Typically the polynomial is fourth order. For





every row a set of polynomial coefficients is calculated. The fitting procedure is executed both for radiance and irradiance measurements. For irradiance measurements the ozone absorption spectrum is excluded from the fit function. The wavelength coefficients $c_{fit,p}(j)$ and the reference column are stored in the L1B calibration output product.

**Table 1: Optical properties for the three channels UV1, UV2 and VIS**

| channel | wavelength range | spectral resolution | spectral sampling | ground pixel size |
|---|---|---|---|---|
| UV1 | 264 – 311 nm | 0.63 nm = 1.9 px | 0.33 nm / px | 13x48 km |
| UV2 | 307 – 383 nm | 0.42 nm = 3.0 px | 0.14 nm / px | 13x24 km |
| VIS | 349 – 504 nm | 0.63 nm = 3.0 px | 0.21 nm / px | 13x24 km |

**Table 2: Standard OMI Level 1 data products**

| OML1BRUG | Global UV Radiance product |
|---|---|
| OML1BRVG | Global VIS Radiance product |
| OML1BRUZ | Spatial Zoom-in UV Radiance product |
| OML1BRVZ | Spatial Zoom-in VIS Radiance product |
| OML1BIRR | Irradiance product |
| OML1BCAL | Calibration product |

**Table 3: Percentage of the RA-affected rows, as of August 2014.**

| channel | UV1 | UV2 | VIS |
|---|---|---|---|
| all orbit phases | 37 % | 33 % | 30 % |
| northern part of orbit phase | 100 % | 57 % | 52 % |

**Table 4: Correction functions for the different measurement types. Generic corrections are applied to all measurement types.**

| Generic | Co-addition division, ADC conversion, Offset correction, Gain Overshoot Correction, Electronic conversion, Non-linearity Correction, binning factor division, Offset calculation, Calculation of measurement noise |
|---|---|





| | Dark current correction | Charge Transfer Efficiency correction | Background correction | Exposure smear correction | Exposure time division | Relative Pixel-to-pixel sensitivity correction | Stray light correction | Slit irregularity correction | Radiance sensitivity conversion | Irradiance sensitivity conversion | Spectral Calibration | Doppler shift correction |
|---|---|---|---|---|---|---|---|---|---|---|---|---|
| Earth | X | x | x | x | x | x | x | x | x | | x | |
| Sun | X | x | x | x | x | x | x | x | | x | x | x |
| WLS | X | x | x | x | x | | x | x | | | | |
| LED | X | x | x | x | x | | | | | | | |
| Dark | | x | | | | | | | | | | |

**Table 5: Solar signal changes observed in the three channels for the timeframe 2005–2015.**

| | UV1 | UV2 | VIS |
|---|---|---|---|
| QVD | 6 % | 4 % | 3 % |
| Regular Al | 3 % | 2 % | 2 % |
| Backup Al | 2.5 % | 2 % | 2 % |

**Table 6: OMI performance over the mission time, year 2005-2015**

| | |
|---|---|
| Dark current | 7 x increase |
| RTS warnings | 0.7 % increase |
| Dead pixel warning | 1.4 % increase |
| Monitoring source LED | 85 % of the original value |
| Monitoring source WLS | 90 % of the original value |
| Gain ratios | ~0.2 % fluctuations |
| Electronic offsets | ~0.5 % fluctuations |
| Non-linearity warning | No significant changes since launch |
| Straylight warning | No significant changes since launch |
| Irradiance decrease | ~6 % QVD diffuser; 3% regular Al; 2.5% backup Al |
| Radiance decrease | ~2.5% UV1; ~1.5% UV2; ~1% VIS |




| Wavelength shifts | ~0.015 nm in UV1; ≤0.005 nm in UV2 and VIS |
|---|---|
| Current status of the Row Anomaly: % of the affected rows | UV2 and VIS: >~30 % (southern hemisphere) and up to ~57 % (part of the northern hemisphere) |
| | UV1: >~37 % (southern hemisphere) and all rows (part of the northern hemisphere) |

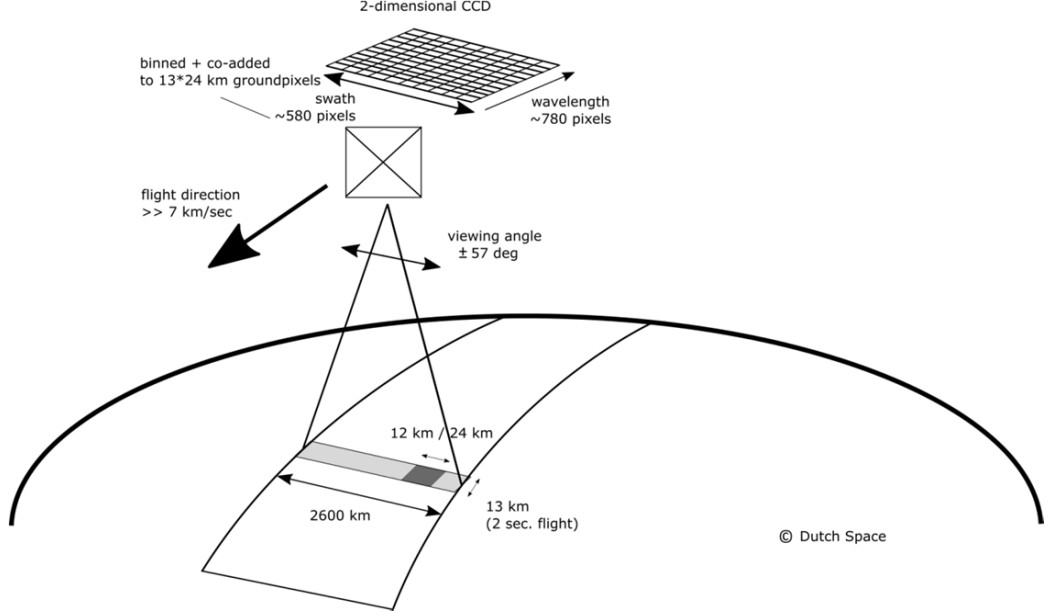

**Figure 1: An impression of OMI flying over the earth globe. The spectrum of a ground pixel is projected on the wavelength dimension of the CCD (the columns). The cross-track ground pixels are projected on the swath dimension of the CCD (the rows). The forward speed of 7 km/sec and an exposure time of 2 seconds lead to a ground pixel size of 13 km in the flight direction. The viewing angle of 114 degrees leads to a swath width on the ground of 2600 km.**




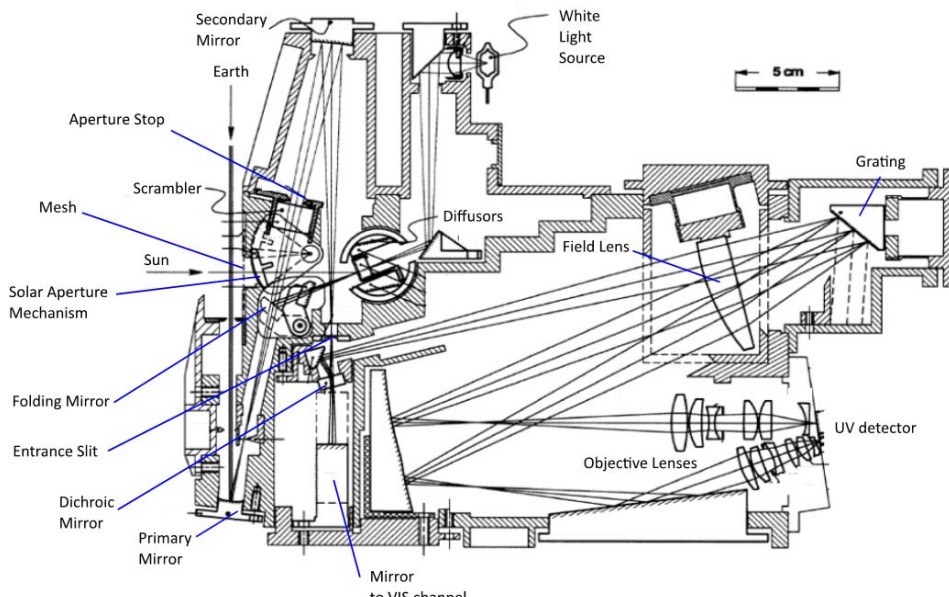

**Figure 2: Optical design of the UV channel. The telescope (Primary and Secondary mirror) is used for both channels. The visible light that passes the dichroic mirror is coupled into the VIS channel. The Folding Mirror is depicted in two positions (earth view and sun / calibration view).**



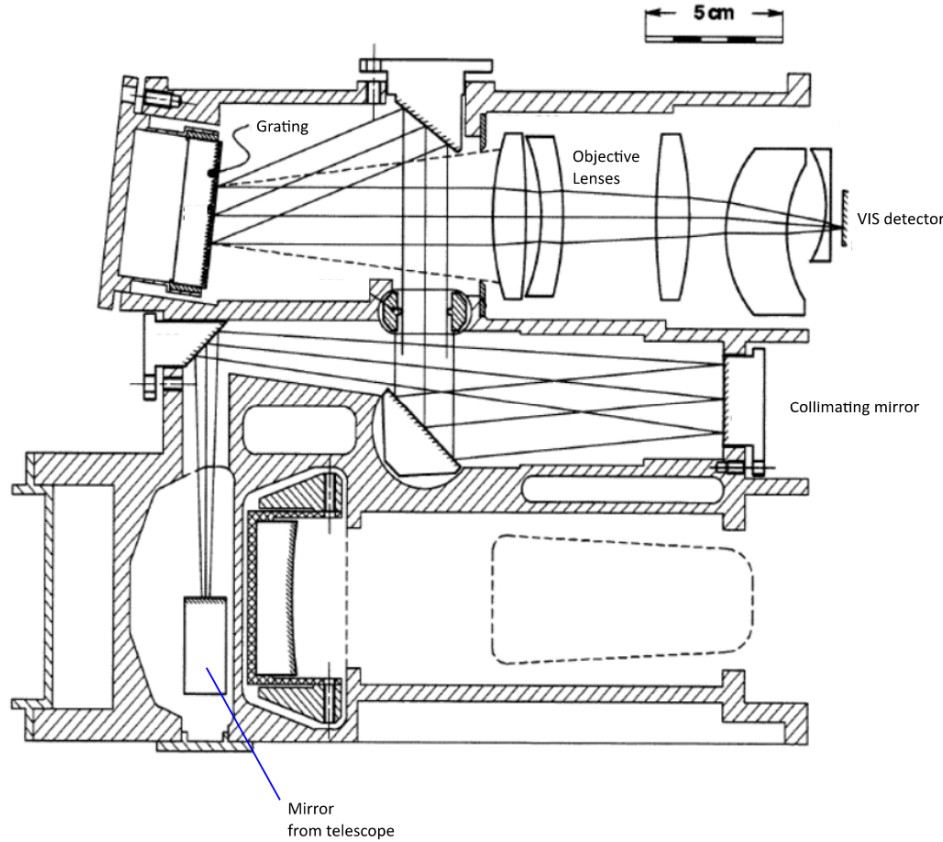

**Figure 3: Optical design of the VIS channel. The light coming from the telescope (not shown) enters the VIS channel via the Mirror from the telescope.**





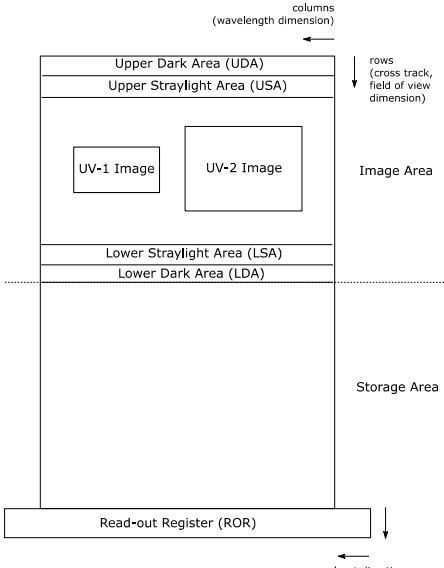

**Figure 4: UV CCD detector layout with two wavelength channels, UV1 and UV2. The VIS CCD detector has similar layout, however with only one wavelength channel.**

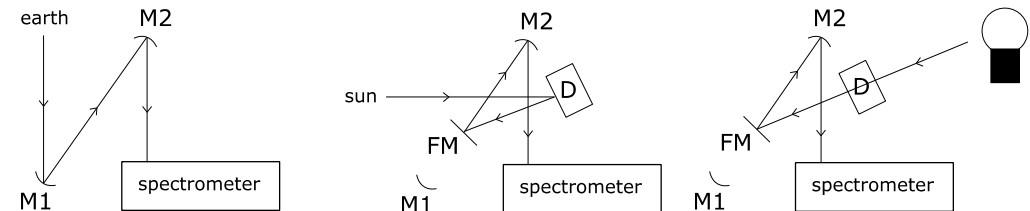

**Figure 5: Schematic optical path. The left panel shows the configuration for earth radiance measurements. M1 and M2 correspond to the primary and secondary mirrors of the telescope. The middle panel shows the configuration for sun irradiance measurements, with the Folding Mirror FM placed between M1 and M2. Sunlight enters the instrument via diffuser D in reflection mode and the FM and M2. The right panel shows the configuration for internal calibration measurements. The light from the White Light Source passes diffuser D in transmission mode and enters the instrument via FM and M2.**





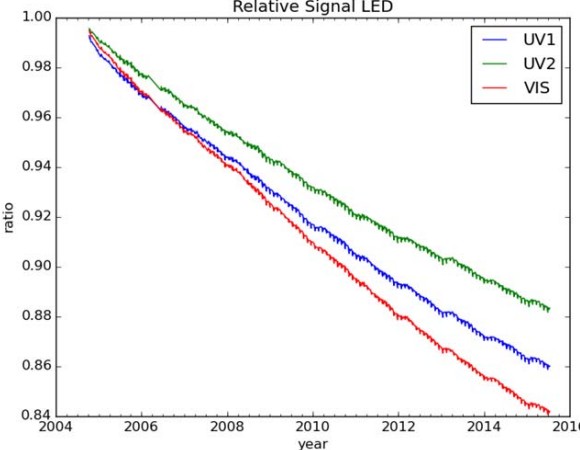

**Figure 6: Signal change of LED during the mission. Each datapoint shows a measurement divided by a reference measurement from the beginning of the mission.**

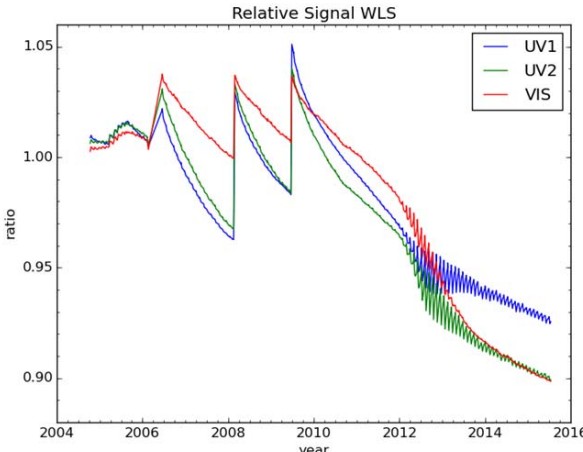

**Figure 7: Signal change of WLS during the mission. Each datapoint shows a measurement divided by a reference measurement from the beginning of the mission. The three abrupt throughput changes in 2006 – 2009 are caused by the long (14 minutes each) WLS duty cycles.**



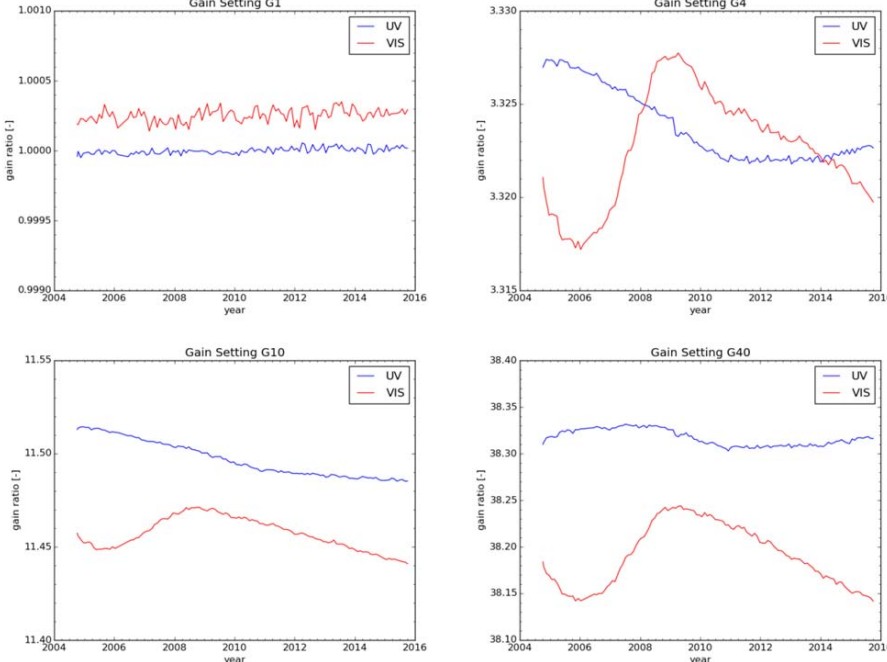

**Figure 8: Gain ratios trends over the mission time for the four gain settings. The very small deviation from 1 for gain setting G1 in the VIS channel (upper left panel) is an indication of the accuracy of the analysis method.**





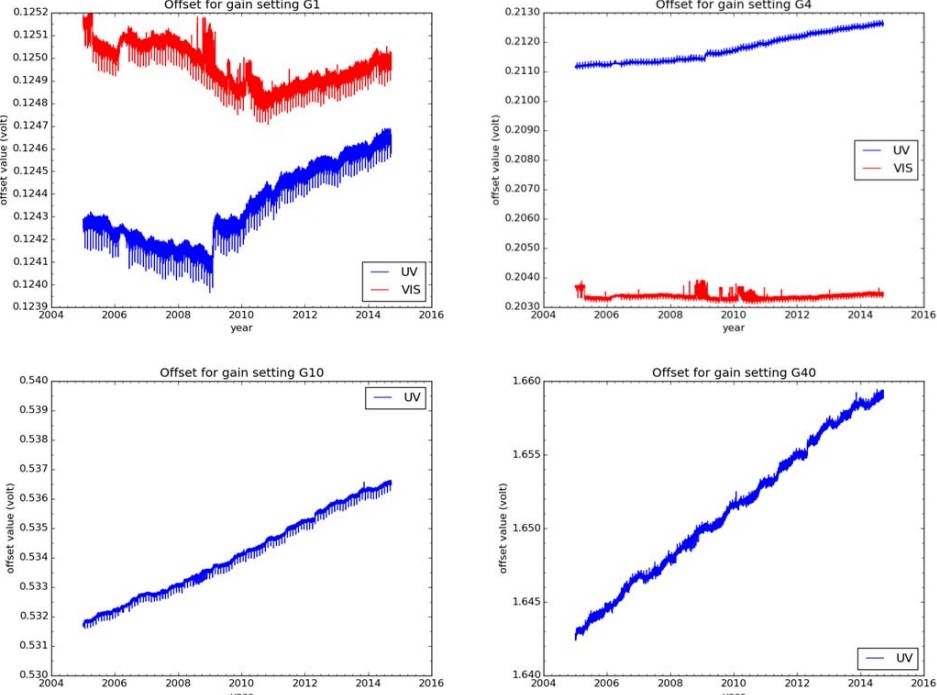

**Figure 9: Changes in the electronic offset over the mission time for the four gain settings. Note that offset values for the gain setting G10 and G40 of the VIS channel are not used in radiance measurements.**

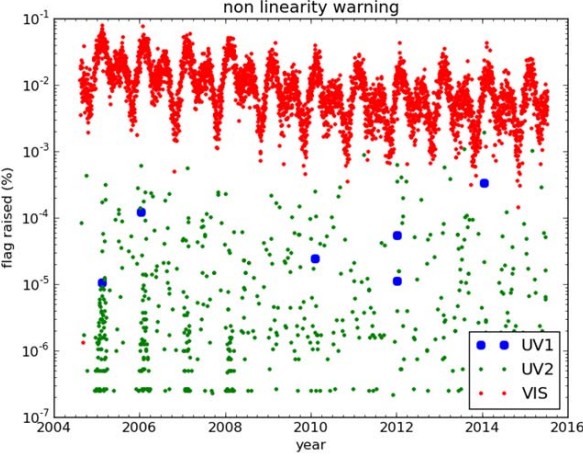





**Figure 10: Non-linearity warnings in output of the CCD amplifiers over the mission time.**

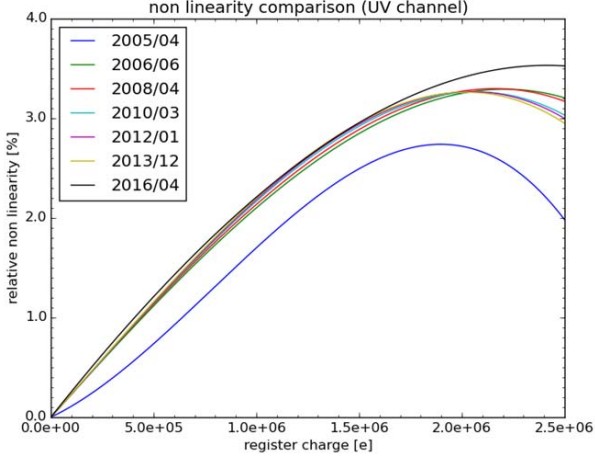

**Figure 11: Non linearity comparison for the UV channel, for different years. Apart from the 2005/04 curve, the curves are pretty**
5    **similar which indicates that non linearity has not changed much during the mission. The curves for the VIS channel are similar to the UV channel curves.**

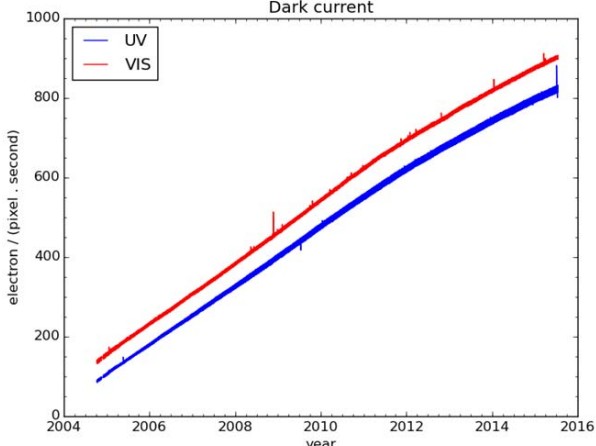

**Figure 12: The average dark currents for two CCDs.**



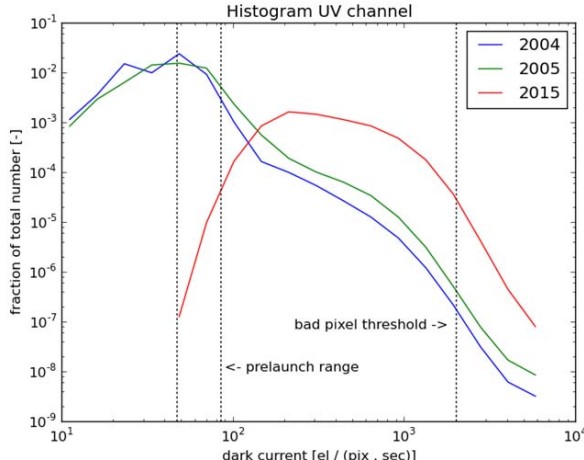

**Figure 13: Histograms of dark current measured in the UV channel.**

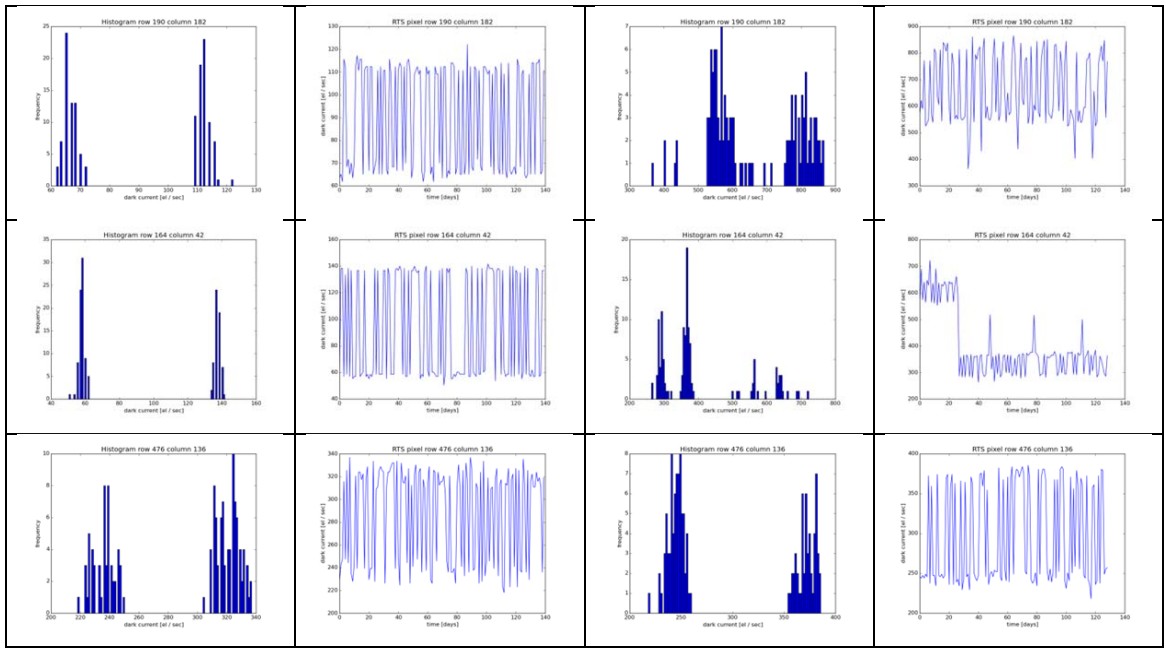

5    **Figure 14: Examples of RTS pixels. Left two columns show results for 2008, right two columns show results for 2015. First and third column show histograms of RTS pixels. Second and fourth column show dark current evolution over time. In 2015 the dark current and noise are higher, which gives the histograms a smoother character. Note the different scales that are used for 2008 and 2015.**



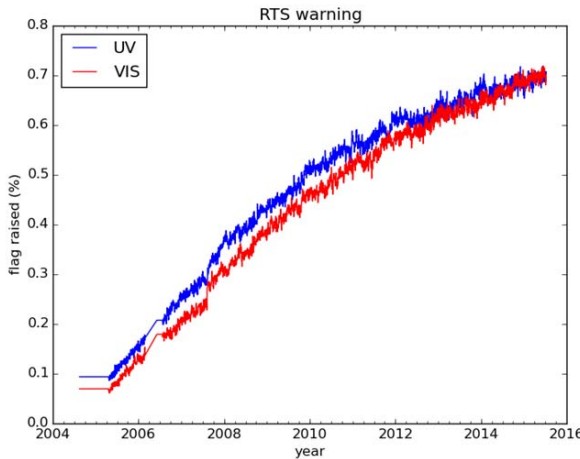

**Figure 15: Random Telegraph Signal (RTS) flagging trend over the mission. These results are for unbinned pixels. The binned L1B pixels have flagging rates that are 8 times higher.**

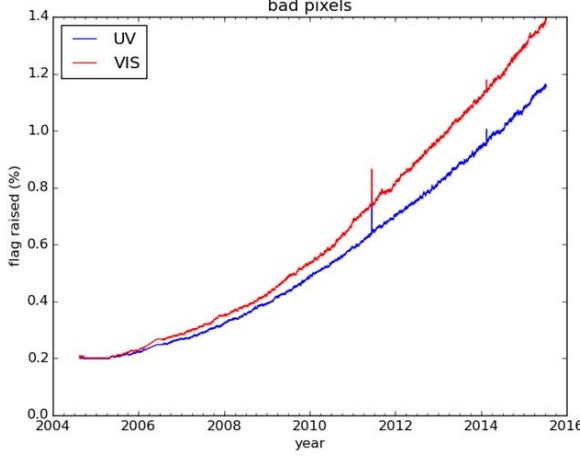

**Figure 16: Bad pixel flagging over the mission time for unbinned pixels. The binned L1B pixels have 8 times higher flagging rate.**





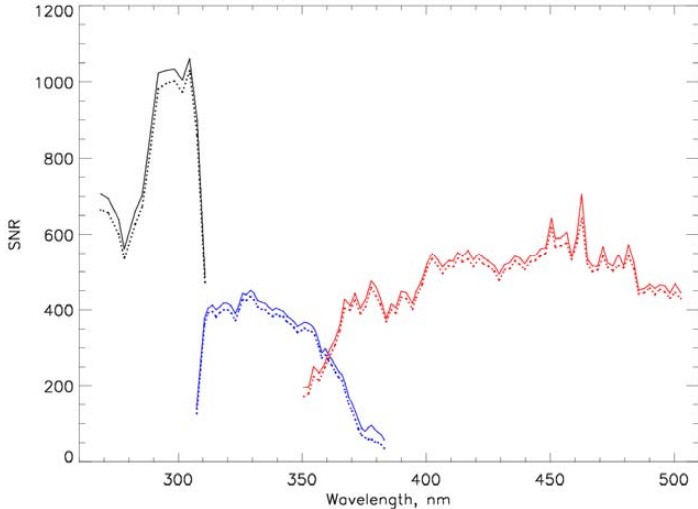

**Figure 17: The wavelength- and time-binned solar irradiance SNRs for January 2005 (full lines) and January 2016 (dots) in UV1 (black), UV2 (blue) and VIS (red) channels.**

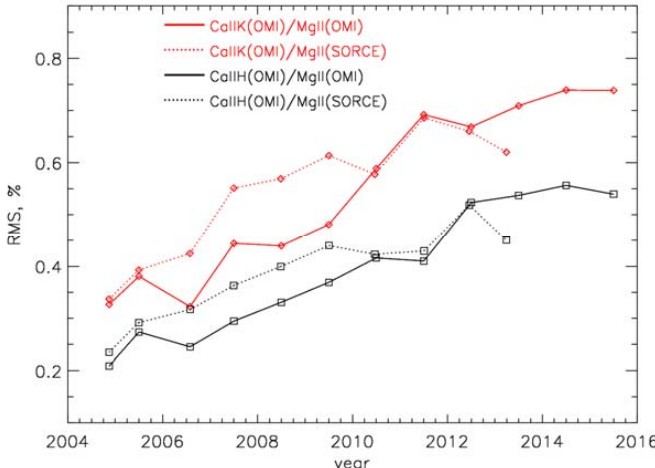

**Figure 18: Time-binned (yearly) RMS of the ratios of the solar indices derived from the VIS and UV1 data.**





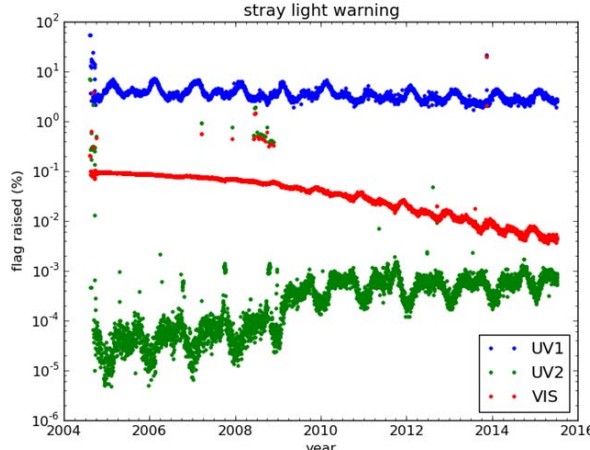

**Figure 19: Stray light warning trend over the mission.**

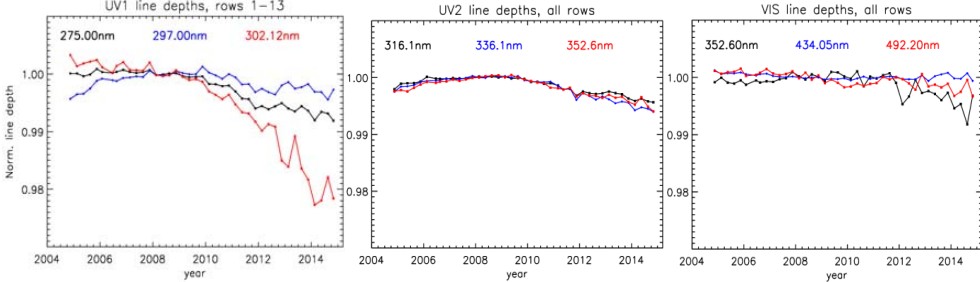

5      **Figure 20: Changes in the normalized line depths of prominent absorption blends in the Solar irradiances observed in UV1 (left), UV2 (center) and VIS (right).**





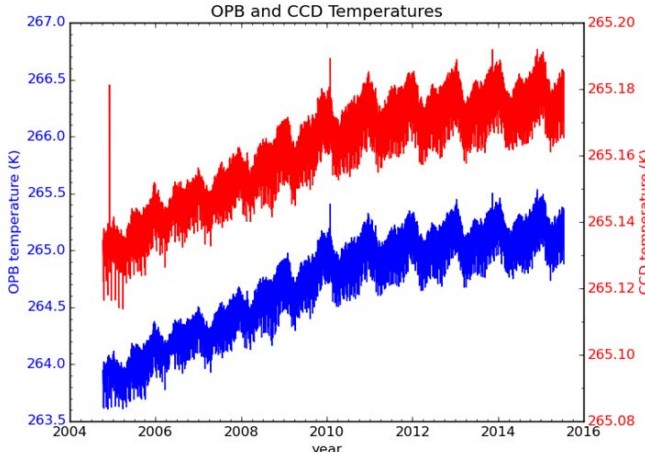

**Figure 21: Changes in the OPB and the UV channel CCD temperatures.**

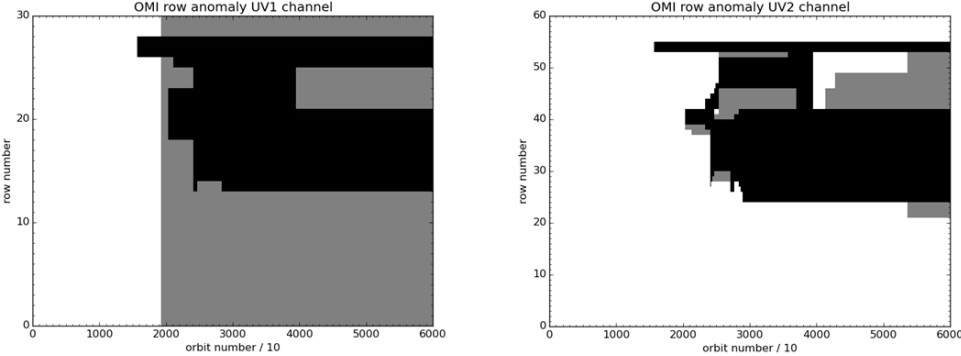

5    **Figure 22: Row anomaly evolution for the UV1 and UV2 channels. Black areas show full RA-affected orbits, grey areas mark partial orbits (northern part). The VIS channel looks similar to the UV2 channel.**

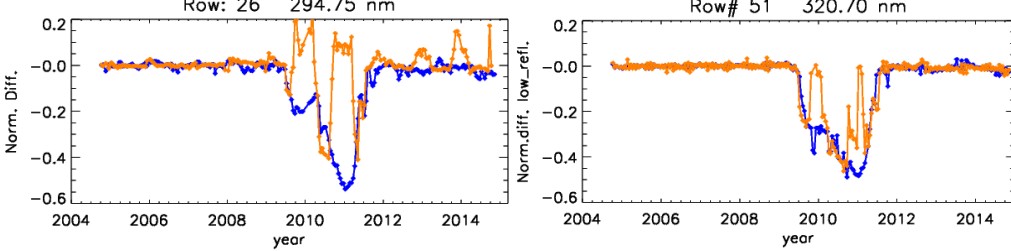




**Figure 23: Left panel shows the wavelength-binned, normalized and de-trended UV1 radiances for the row #26 (counting from 1) for the 30-degree latitudinal bins centered on lat=-45 deg (blue) and lat=45 deg (orange). Right panel shows the same for the UV2 low-reflectivity (r < 15%) sub-sample of radiances registered by the row #51 (in correspondence to the FOV of the UV1 row #26) . The plotting ranges correspond to -60%/+20% changes of radiances.**

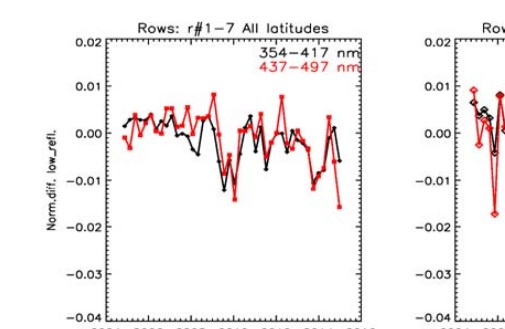
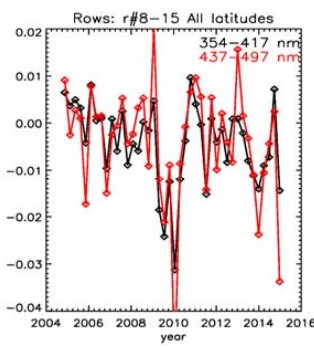
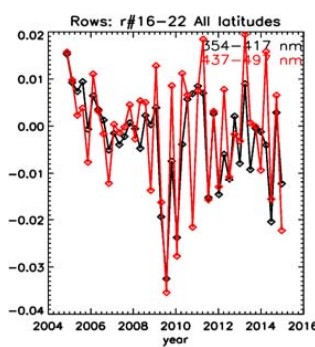

**Figure 24: The time-, row- and wavelength-binned, normalized and seasonally de-trended VIS radiances sampled at all latitudes over the low-reflectance (r<10%) terrain. All shown rows are presumably not affected by the row anomaly; though notice the changes around 2009-2010.**

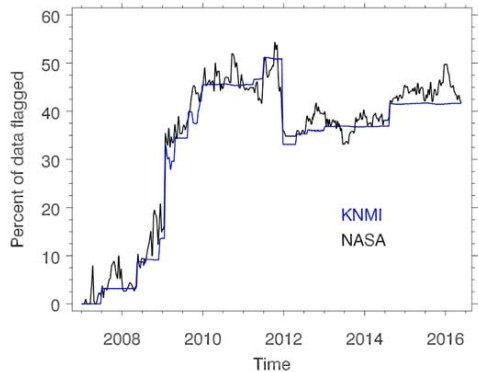

**Figure 25: Percent of data flagged by the two row anomaly flagging algorithms in use for the UV2 channel. The KNMI algorithm is used to flag the L1B radiance products while the NASA algorithm used in several NASA retrieval algorithms to flag the L2 data. Though the physical basis of the two algorithms is rather different, they produce consistent flagging results over the full course of the OMI mission. The presence of high frequency variations in the NASA flagging algorithm is due to the fact that it flags data dynamically, while the KNMI row anomaly flags are changed as need determines.**





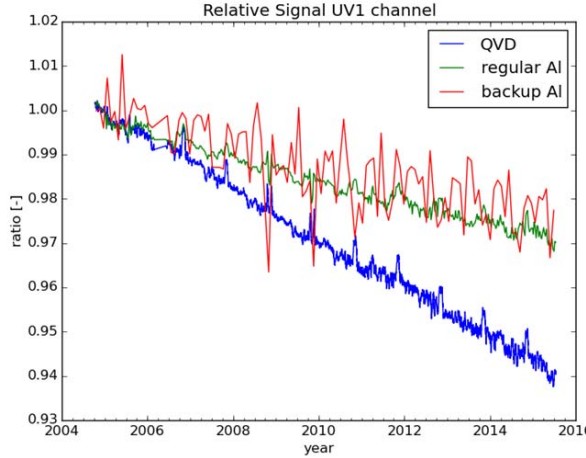

**Figure 26: Relative solar signal of the UV1 channel over the mission. Each data point shown for the 3 diffusers is a result of a spectral and spatial average over the entire channel. The higher rate of signal change from the frequently used QVD suggests degradation in diffuser reflectivity caused by solar exposure.**

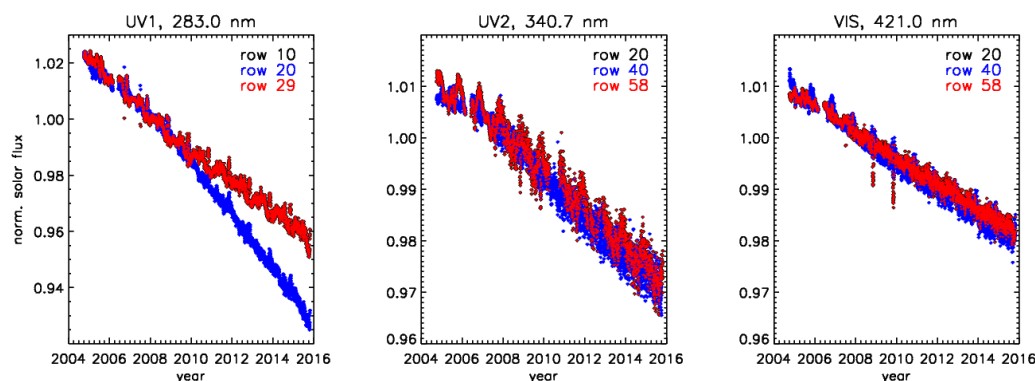

**Figure 27: The wavelength-binned and normalized QVD solar irradiances for different rows in the UV1 (left), UV2 (center) and VIS (right panel) channels. The increase in change with decreasing wavelength is typical of optical degradation related to solar exposure. The anomalous change in the UV1 row 20 beginning in 2009 may be caused by additional solar exposure resulting from the row anomaly reflections.**





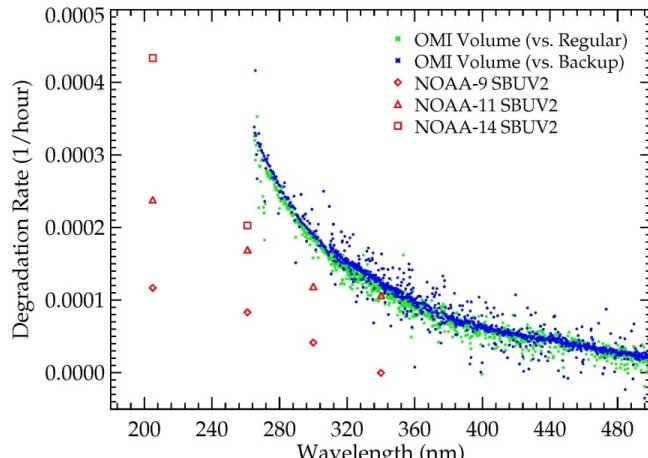

**Figure 28: The fractional change in the QVD diffuser per hour of solar exposure relative to the regular and backup Al diffuser**





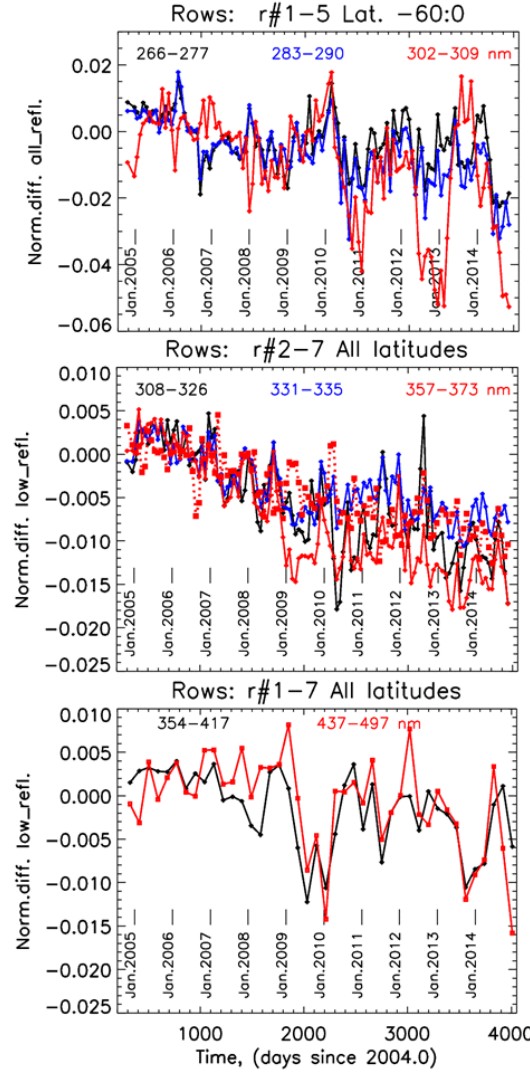

**Figure 29: Top panel: the de-trended, normalized time-, wavelength-, latitude- and row-binned UV1 radiances. Central panel: the same for UV2 for the low-reflectivity (<10%) subsample (full red line) and the high-reflectivity (>80%) data (dotted red line) for the 357-373 nm range. Bottom panel: the low-reflectivity, binned radiance trends for VIS. Note the change of the y-axis plotting scales for different OMI channels.**



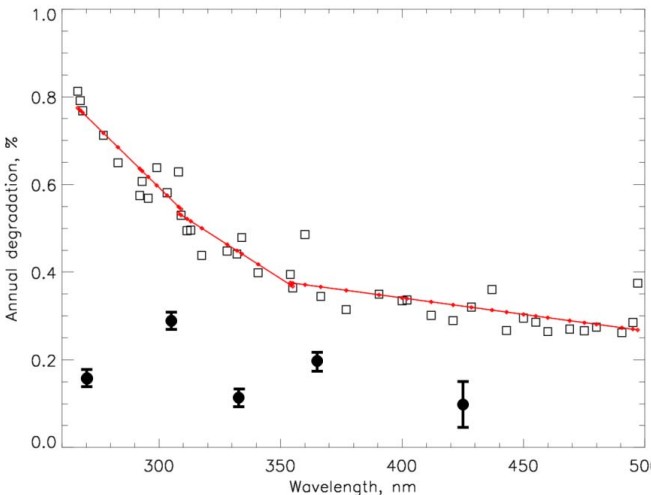

**Figure 30: The annual signal change rates (QVD diffusor, squares) derived from the wavelength-binned Solar irradiances for 2007-2009 (see more details in Marchenko and Deland (2014)) and fitted with linear polynomials (red lines) for each OMI channel. Filled circles and ±σ error bars show the degradation estimates in the OMI radiances.**

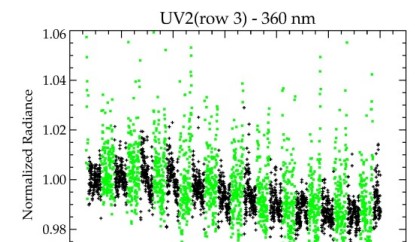
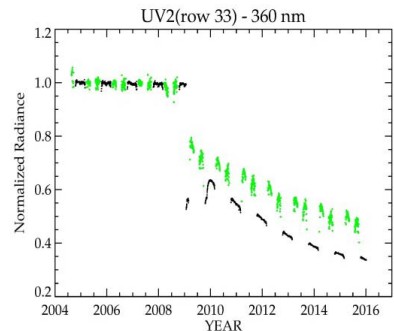

**Figure 31: Measured daily mean radiances in UV2 at 360 nm over Antarctica (black) and Greenland (green) are shown as a function of time relative to initial measurements in 2004. Left panel shows results in Row 3, which is at the western edge of the OMI swath and is far from the row anomaly-related blockage. Right panel shows results for Row 33, which is near nadir and is affected by RA.**




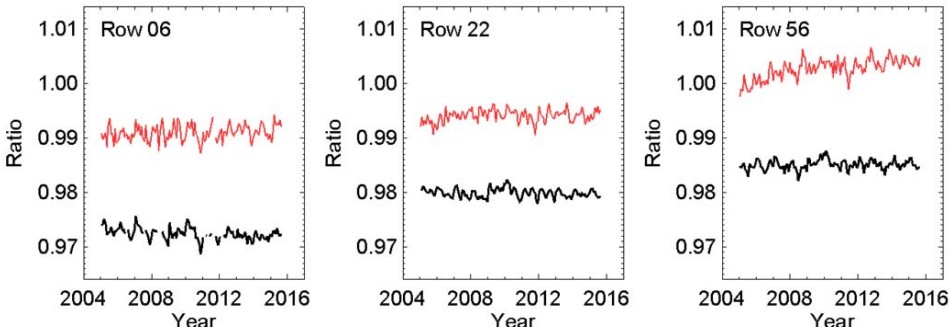

**Figure 32: Trends in monthly mean ratios of OMI radiances measured at 340 and 354 nm on the UV-2 detector (black) and 354 and 380 nm on the VIS detector (red) for three detector rows. The observed trend in these Earth-view measurement ratios is < 0.5% per decade in both channels. Detector rows 6, 22, and 56 shown here are among those which are unaffected by the row anomaly. The data shown here were selected to minimize geophysical effects of wavelength dependence and measurement-geometry on the radiance ratios. Conditions are limited to high reflectance ($\rho > 0.9$), low solar zenith angle ($\theta_0 < 50°$) over the tropical Pacific (20N-S, 130-170W), where radiation is scattered primarily by high altitude, deep convective clouds and the observed atmosphere is generally free of aerosol. The more noticeable trend in row 56 is not understood. Despite specifically selecting data to minimize wavelength dependent effects, some residual geophysical variation remains in the ratios. These are most likely due to cloud scattering phase function variation with cloud water phase and from residual multiple scattering effects varying seasonally over the SZA range of the measurements.**

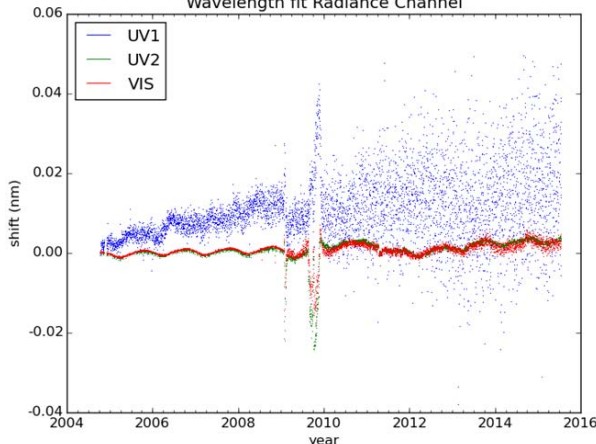

**Figure 33: Spectral calibration trend over the mission for the radiance channel. Spectral calibration for the UV1 channel is strongly affected by the row anomaly effect since 2009.**




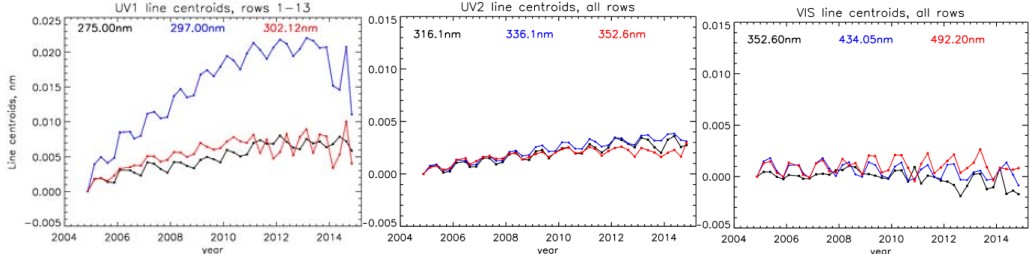

**Figure 34: Changes in the line centroids of prominent absorption blends in the Solar irradiances observed in UV1 (left), UV2 (center) and VIS (right).**

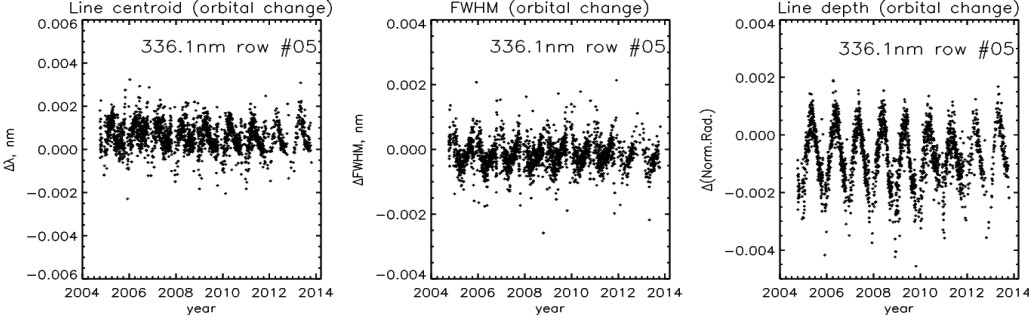

**Figure 35: The orbital changes of the line centroids (left), line FWHMs (center) and line depths (right) in the UV2 radiances for the spectral blend at 336.1nm and the row#05.**