# Peer review of "In-flight performance of the Ozone Monitoring Instrument"

_Atmospheric Measurement Techniques, 2016_

## Referee Comment (RC1) · M. Dobber (Referee) · 17 Jan 2017

This paper had previously been submitted for Atmospheric Chemistry and Physics (ACP), now for Atmospheric Measurement Techniques (AMT). My review comments were already provided for the previous ACP submission, and accounted for by the authors in this updated manuscript, hence I have no further comments, hence I propose to publish as is.

---

## Referee Comment (RC2) · Anonymous Referee #2 · 21 Feb 2017

This manuscript presents a comprehensive overview of the Ozone Monitoring Instrument, the calibration procedures, the instrumental performance and its evolution in time. With the notable exception of the row anomaly, OMI is a very stable instrument and this paper documents well the various components of the calibration chain and their respective performance. It is generally well written and illustrated by a large number of figures facilitating the reading. As the first referee said, the manuscript has already been consolidated before. I have a limited number of comments and I recommend the publication of this manuscript within AMT once they have been considered.

**Comments**

- In L2 retrieval algorithms using OMI observations, it is often recommended to use consolidated static solar spectra (taken at the beginning of the mission) as the refer-

ence. This is not addressed at all in the manuscript. Could you add this information in the manuscript where it fits the best, explain how these consolidated sun spectra have been constructed and what are the actual limitations with the daily recorded sun spectra and if there are some specific conditions where it would be profitable to use them anyway.

- For the in-flight stray light characterization based on the monitoring of the position and shapes of a few isolated absorption lines in the Earthshine spectra, can inelastic scattering (Ring effect) perturb the procedure? Inelastic scattering tends to reduce the line depth, similarly to presence of stray light.

- Appendix A: In eq. (A.12), what do the scaling parameters *sf* represent? Is it not somewhat redundant with the fit parameters $a_n$? Please clarify the physical meaning of these parameters. Also, you should specify that the reference spectra (O3 and Ring cross-sections, reference sun spectrum) need to be pre-convolved at the instrumental resolution before the wavelength calibration fit. Or is there some kind of fit of the slit function during the wavelength calibration itself? Have you monitored the possible time evolution of the instrumental slit functions (depending on the spectral range and row)?

- In table 1, for the UV1 channel, the spectral sampling appears to be not sufficient for the corresponding spectral resolution (only 1.9 pxl for the specified FWMH). Does it mean that the recorded spectra are undersampled? What are the implications for L2 products?

- In the original version of the manuscript submitted to ACPD, there was a section making the link with L2 retrievals. Unfortunately, this section has been removed. I would recommend to put it back as the motivation of having well characterized and calibrated OMI spectra is obviously to generate afterwards L2 geophysical products of very high quality (which is indeed the case with OMI).

**Technical comments**

- Page 1 - lines 25-26: Capitalize 'SCIAMACHY' and 'EUMETSAT'.

- Page 2 - line 19: replace "it's" by "its" (two occurences)

- Page 4 - line 5: "The QVD diffuser is used"

- Page 7 - line 21: add "(RTS)" after "random telegraph signal"

- Page 7 - line 30: Specify which gain is assigned to which part of the spectrum (spectral range).

- Page 10 - line 3: "This analysis has been done" and not "This analysis is been done"

- Page 11 - line 19: define "SORCE"

- Page 12 - Figure 19: in the figure legend, I think there is an inversion between green and red for UV2 and VIS.

- Page 19 - line 7: Sentence truncated?

- Page 19 - line 19: "lest uncertainties"?? - Sentence not clear

- Page 21 - line 17: "These variations are most ??" - add "likely" after "most".

- Page 26 - line 21: "data rate" instead of "datarate"

- Page 29 - line 2: "For irradiance measurements, the ozone and Ring absorption spectra are excluded. . ."

- Figure 25: add "is" after "while the NASA algorithm"

---

## Author Response (AR1)

Comments

Comment 1

- In L2 retrieval algorithms using OMI observations, it is often recommended to use consolidated static solar spectra (taken at the beginning of the mission) as the reference. This is not addressed at all in the manuscript. Could you add this information in the manuscript where it fits the best, explain how these consolidated sun spectra have been constructed and what are the actual limitations with the daily recorded sun spectra and if there are some specific conditions where it would be profitable to use them anyway.

Answer: we will do it.

New lines: p22, line 33:

Many trace-gas retrieval algorithms rely on solar reference spectra, thus creating additional dilemma: should the referencing use static (usually, chosen at the beginning of the mission) or dynamic solar data? Choosing between two approaches, one should take into consideration a multitude of conflicting requirements. Among them is the possible different degradation rates in the optical channels acquiring radiance and irradiance data. OMI shows such differences, though they are relatively small. Also, one has to consider the gradual SNR decrease in the solar data (cf. Figure 18). Even in the moderate-resolution OMI irradiances, the daily, monthly, as well as long-term solar variability is prominent in the lambda < 450 nm domain, frequently exceeding 0.5% in the strong spectral blends (Marchenko and DeLand, 2014), calling for a thorough evaluation of sensitivity of the L2 science products to the variable Solar spectrum. On the other hand, any substantial (far exceeding the sensitivity of a typical trace-gas retrieval algorithm) long-term wavelength drifts may require extensive interpolations of the static solar data, thus augmenting the under-sampling biases (Kurosu et al., 2004).

The frequently used in the OMI L2 applications static Solar spectrum was produced (T. Kelly, priv. comm. 2017) from seven subsequent daily solar observations acquired between December 28, 2004 - January 3, 2005. This static reference spectrum is derived as an unweighted average of the daily observations censured for exceedingly large deviations from the corresponding row (i.e., FOV)- and wavelength-dependent median values.

Comment 2

- For the in-flight stray light characterization based on the monitoring of the position and shapes of a few isolated absorption lines in the Earthshine spectra, can inelastic scattering (Ring effect) perturb the procedure? Inelastic scattering tends to reduce the line depth, similarly to presence of stray light.

Answer:

We have already mentioned this in the text. In p.22, line 26 we note: "The line-depth variability show clear ±0.2% seasonal fluctuations, most likely related to changes in the Ring line-filling factors, with their direct proportionality to the seasonably changing (Solar elevation for a given latitude) atmospheric path-lengths."

Comment 3

- Appendix A: In eq. (A.12), what do the scaling parameters sf represent? Is it not somewhat redundant with the fit parameters an? Please clarify the physical meaning of these parameters. Also, you should specify that the reference spectra (O3 and Ring cross-sections, reference sun spectrum) need to be pre-convolved at the instrumental resolution before the wavelength calibration fit. Or is there some kind of fit of the slit function during the wavelength calibration itself? Have you monitored the possible time evolution of the instrumental slit functions (depending on the spectral range and row)?

Answer: we will explain it

New lines: p 28, line 4:

The sun spectrum, ozone absorption spectrum and Ring spectrum need to be pre-convolved with the OMI Instrument Spectral Response Function. The scaling parameters $sf_{DOAS,n}$ are used to keep the fit parameters $a_n$ in a range in which the derivative calculation during the fit process provides optimal performance.

Answer: monitoring of the instrumental slitfunction was described in p 22 lines 20-32:

Long-term (mission time) and short-term (orbital) stability of the instrument spectral response function is deemed important for reliable, unbiased retrievals of the atmospheric trace-gas properties. Changes in the instrument spectral response affect depths and widths of the detected spectral features. In Figure 35 we show variations of the line-profile parameters derived from radiances for the line blend around λ=336.1 nm and the UV2 row#5. Each panel shows the differences between the latitude- and time-binned early-orbit (lat=-60 to -50) and late-orbit (lat=40 to 50) line-profile parameters. The orbit-differentiated wavelength registration and FWHMs go through relatively minor (±0.001 nm) seasonal changes which we deem negligible in comparison to the 0.14 nm UV2 sampling rate. The line-depth variability show clear ±0.2% seasonal fluctuations, most likely related to changes in the Ring line-filling factors, with their direct proportionality to the seasonably changing (Solar elevation for a given latitude) atmospheric path-lengths. The line centroids are also involved in ±0.001 nm seasonal cycling. However, such fluctuations should be regarded as negligible in comparison to the 0.142 nm sampling rate in the UV2 spectra. The shown trends are representative for all rows. Hence, we may conclude that instrumental factors do not introduce observable (i.e., exceeding our sensitivity limits) spectral response changes along the OMI orbit. Nor such factors cause any long-term (mission time) instrumental trends exceeding ~0.2% in measurements of the UV2 and VIS absorption features (see Figure 20).

Comment 4

- In table 1, for the UV1 channel, the spectral sampling appears to be not sufficient for the corresponding spectral resolution (only 1.9 pxl for the specified FWMH). Does it mean that the recorded spectra are undersampled? What are the implications for L2 products?

Answer: we will clarify this

New lines: p 2, line 23:

For the UV2 and VIS channel the spectral sampling is 3 pixels for the Full-Width Halve Maximum (FWHM). For the UV1 channel this is 1.9 pixel for the FWHM, which implies that the UV1 channel is undersampled. This is not a problem for operational use of OMI, because the UV1 channel is mainly used for ozone profile retrieval, which uses absolute radiances, and does not rely on spectral fitting.

Comment 5

- In the original version of the manuscript submitted to ACPD, there was a section making the link with L2 retrievals. Unfortunately, this section has been removed. I would recommend to put it back as the motivation of having well characterized and calibrated OMI spectra is obviously to generate afterwards L2 geophysical products of very high quality (which is indeed the case with OMI).

Answer: we will take the contents of this section, and put it in different places in the article where it fits best.

New lines:

P 3, line 4:

The quality of information in Level 1 data products is a somewhat relative concern from the standpoint of different retrieval applications, since there is considerable variety in the sensitivity of different retrievals to errors and instrument degradation in the Level 1 data. A full review of these sensitivities is beyond the present scope of this paper, but, where appropriate we summarize them for additional context.

P 10, line 17:

Despite the increase observed in RTS on the OMI CCD detectors, the short-lived nature of these events appears to limit their overall impact on the most sensitive Level 2 retrievals. For example in the OMI BrO spectral fitting algorithm (Kurosu et al., 2004), the fitting residuals used for diagnostic purposes grow by less than 5% over the OMI mission (Kurosu, personal communication). The same applies to the fitting residuals of the OMCLDRR fitting algorithm (ref. [26]) (Vasilkov, personal communication).

P 12, line 23:

Science products that derive information from the discrete intensity or ratio of reflectances are more sensitive to stray light errors, while the DOAS, spectral fitting, and PCA algorithms are relatively insensitive to stray light.

P 21, line 26:

Level 2 retrievals which use Differential Optical Absorption Spectroscopy (DOAS) or Principal Components Analysis (PCA) techniques to derive trace gas information from the high spectral frequency structure in Level 1 reflectance measurements can be sensitive to wavelength errors as small as 1/100th of the OMI wavelength sampling interval (refs. [11] and [19]).

Technical comments

TC1

- Page 1 - lines 25-26: Capitalize 'SCIAMACHY' and 'EUMETSAT'.

Answer: we will do it.

TC2

- Page 2 - line 19: replace "it's" by "its" (two occurences)

Answer: we will do it.

TC3

- Page 4 - line 5: "The QVD diffuser is used"

Answer: we will do it.

TC4

- Page 7 - line 21: add "(RTS)" after "random telegraph signal"

Answer: we will do it.

TC5

- Page 7 - line 30: Specify which gain is assigned to which part of the spectrum (spectral

range).

Answer: we will provide the information.

New lines: p 8, line 3:

The gain values for the different channels and spectral bands is shown in Table 3.

**Table 3: Electronic gain values for different channels and spectral bands**

| channel | λ min (nm) | λ max (nm) | gain value |
|---|---|---|---|
| UV1 | 264 | 286,2 | 40 |
| | 286,2 | 301,5 | 10 |
| | 301,5 | 311 | 1 |
| UV2 | 307 | 383 | 1 |
| VIS | 349 | 358,8 | 4 |
| | 358,8 | 504 | 1 |

TC6

- Page 10 - line 3: "This analysis has been done" and not "This analysis is been done"

Answer: we will do it.

TC7

- Page 11 - line 19: define "SORCE"

Answer: we will do it.

New line:

SOlar Radiation and Climate Experiment (SORCE)

TC8

- Page 12 - Figure 19: in the figure legend, I think there is an inversion between green and red for UV2 and VIS.

Answer: We agree that it seems strange that the red line (VIS channel) is above the green line (UV2 channel). But we have checked the data, and this figure shows the correct result.

TC9

- Page 19 - line 7: Sentence truncated?

Answer: we will update the line

New line:

We can isolate the optical degradation of the solar diffusers by comparing the signal changes observed with each one. Figure 28 shows the fractional change in the QVD per hour of solar exposure relative to the other two diffusers.

TC10

- Page 19 - line 19: "lest uncertainties"?? - Sentence not clear

Answer: we will update the line

New line:

Still, these results should serve as a warning to minimize the exposure of the least-used diffuser for fear that uncertainties in its degradation rate become a significant component in the calibration error budget.

TC11

- Page 21 - line 17: "These variations are most ??" - add "likely" after "most".

Answer: we will do it.

TC12

- Page 26 - line 21: "data rate" instead of "datarate"

Answer: we will do it.

TC13

- Page 29 - line 2: "For irradiance measurements, the ozone and Ring absorption

spectra are excluded…"

Answer: we will do it.

TC14

- Figure 25: add "is" after "while the NASA algorithm"

Answer: we will do it.

[revised manuscript text omitted]